# UniPruning: Unifying Local Metric and Global Feedback for Scalable Sparse LLMs

## Abstract

Large Language Models (LLMs) achieve strong performance across diverse tasks but face prohibitive computational and memory costs. Pruning offers a promising path by inducing sparsity while preserving architectural flexibility. However, existing methods struggle to balance efficiency and robustness: local metric approaches prune layer by layer but often collapse under high sparsity, whereas global feedback methods enforce consistency at the cost of expensive weight updates or restrictive semi-structured formats. We present **UniPruning**, a unified post-training pruning framework that combines the speed of local saliency metrics with the stability of global coordination, enabled by a mirror descent based optimization, all **without updating model weights**. UniPruning leverages fast layer-wise scoring and a lightweight global controller to allocate a single sparsity budget, supporting both unstructured and semi-structured $N{:}M$ pruning within one framework. After a brief calibration, it can generate pruning masks for arbitrary sparsity levels in one shot, and adapts seamlessly to hardware-aware constraints. Extensive experiments on multiple pretrained LLM families and standard benchmarks show that UniPruning consistently delivers competitive or superior perplexity and zero-shot accuracy. Ablation studies further highlight the importance of mirror descent and local saliency anchoring. Overall, UniPruning provides an efficient, principled, and scalable solution for sparsifying large-scale LLMs.

## 1 Introduction

Large Language Models (LLMs) (Achiam et al., 2023; Touvron et al., 2023; Zhang et al., 2022) have redefined the frontier of natural language processing, achieving unprecedented capabilities across diverse tasks. Yet, their deployment at scale remains constrained by prohibitive computational and memory costs driven by their enormous parameter counts. To bridge this gap, model compression has emerged as a critical direction, with quantization (Lin et al., 2024), distillation (Gou et al., 2021), and pruning (Frantar & Alistarh, 2023) as key strategies. Among these, pruning stands out for its ability to induce sparsity while preserving architectural flexibility, thereby delivering substantial reductions in both memory footprint and computational demand.

Existing pruning paradigms for LLMs differ along two axes: **structural granularity and algorithmic coordination**. Structurally, pruning ranges from *unstructured pruning*, which removes individual weights for fine-grained control but suffers from limited hardware acceleration, to *structured pruning*, which eliminates entire channels or neurons to enable efficient execution on modern accelerators. *Semi-structured pruning* (Mishra et al., 2021), such as the widely adopted $N{:}M$ format, strikes a practical balance, enabling substantial sparsity with hardware-friendly patterns.

From an algorithmic perspective, pruning methods fall into two categories: local metric and global feedback. Local approaches, such as SparseGPT (Frantar & Alistarh, 2023) and Wanda (Sun et al., 2024), make layer-wise pruning decisions based on weight and activation statistics, offering simplicity but often failing under high sparsity due to ignored cross-layer dependencies. Global feedback methods address this by introducing model-wide coordination through regularization or mask learning, as seen in SparseLLM (Bai et al., 2024) and ProxSparse (Liu et al., 2025). While more consistent, these approaches can be computationally costly or restricted by specific sparsity formats.

In this paper, we introduce **UniPruning**, a unified pruning framework that combines the speed of local metrics with the consistency of global feedback, all without requiring weight updates. To integrate these two objectives, we adopt the mirror descent algorithm as a principled approach for joint optimization. UniPruning employs a fast, layer-wise scoring step to extract local evidence, coupled with a lightweight global controller that redistributes a single sparsity budget across layers using a mirror-descent projection (Beck & Teboulle, 2003; Nemirovsky & Yudin, 1983). Concretely, model weights evolve along a gradient flow while an auxiliary saliency variable $\Gamma$ is updated under a sparsity-aware projection. This mechanism naturally supports both unstructured and semi-structured ($N{:}M$) pruning within one framework. After calibration, masks are generated by a single sorting operation on $\Gamma$ and directly applied to the original pretrained weights, enabling one-shot extraction of multiple sparsity levels. To stabilize pruning decisions, UniPruning incorporates local saliency signals from a calibration set as robust local signals, while the global controller ensures balanced allocation across layers. This synergy yields pruning that is as efficient as local methods, yet globally consistent and structurally aware like feedback-based approaches.

To evaluate the effectiveness of our method, we conduct extensive experiments across a diverse set of large language models. We benchmark UniPruning under both unstructured and semi-structured sparsity regimes, comparing it against widely-used post-training pruning baselines. Our results show that UniPruning consistently achieves competitive or superior performance in terms of perplexity and zero-shot accuracy, especially under high sparsity levels. Notably, it maintains model stability where other methods degrade, and achieves strong average results across models and tasks—all while avoiding any weight updates during pruning. We also perform detailed ablation studies to validate the role of mirror descent and the choice of local saliency metrics. Our contributions are:

- **A unified view of local metric and global feedback pruning.** UniPruning offers a framework that keeps the simplicity of local metric, layer-wise evidence while coordinating a model-wide sparsity budget through a global regularizer and one-shot ranking. This unification maintains layer-level structure preservation and improves whole-model trade-offs under a common budget.

- **Mirror-descent pruning without weight update.** We extend mirror descent to LLM pruning by learning a saliency variable $\Gamma$ jointly with weights and anchoring it to data-driven local saliency metrics. The same procedure supports both unstructured sparsity and semi-structured patterns. By avoiding weight updates and relying solely on learned saliency, the method remains lightweight, preserves accuracy, and is practical for scaling to large language models.

- **Extensive evaluation across models and sparsity.** We test UniPruning on multiple pretrained LLMs and standard benchmarks, comparing against previous state-of-the-art pruning baselines. The method consistently delivers strong accuracy at moderate-to-high sparsity while remaining efficient. Our results indicate that mirror-descent saliency, anchored to local metric, is a robust drop-in route for both unstructured and semi-structured ($N{:}M$) LLM sparsification.

## 2 RELATED WORK

**LLM pruning.** The tension between local efficiency and global coordination becomes even more pronounced when scaling pruning to large language models. On the one hand, lightweight local approaches such as Wanda (Sun et al., 2024), RIA (Zhang et al., 2024), stochRIA (Yi & Richtárik, 2025) show that activation-aware (relative activation-aware) scoring can prune both unstructured and semi-structured ($N{:}M$) patterns effectively, requiring only a small calibration set and minimal computation. On the other hand, methods like SparseGPT (Frantar & Alistarh, 2023) push pruning into the LLM regime by incorporating approximate second-order information: they prune many weights in one shot and apply a local least-squares correction to stabilize outputs. These advances confirm that post-training pruning can achieve strong efficiency–accuracy trade-offs at LLM scale, even without retraining. However, they remain fundamentally layer-local: each layer is pruned largely in isolation, with limited capacity to balance sparsity across the entire model. This gap highlights the central open question for LLM pruning: how to unify the speed and practicality of local methods with the robustness and balance of global coordination, especially under extreme sparsity or hardware-specific formats.

**Mirror descent.** Mirror descent (Nemirovsky & Yudin, 1983; Beck & Teboulle, 2003; Bubeck, 2015; Ding et al., 2025) is a general framework for constrained, geometry-aware optimization. It maps parameters into a dual space, takes gradient steps there, and projects back via a mirror map

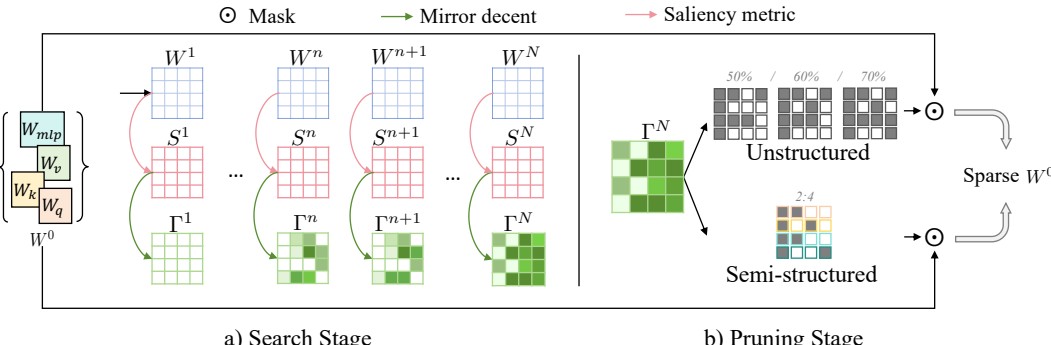

a) Search Stage          b) Pruning Stage

Figure 1: Overall framework of **Unified Pruning**. The framework targets pruning in two types of layers: **MLP layers** and **attention projection layers**. It operates in two stages (a) **Search Stage:** model weights $W$ are iteratively updated while saliency variables $\Gamma$ are jointly optimized with local metrics $S(W)$ via mirror descent, gradually accumulating pruning signals. (b) **Pruning Stage:** the final $\Gamma^N$ is projected into unstructured or semi-structured sparsity masks, which are applied to the original pretrained weights $W^0$ to yield sparse models at arbitrary sparsity levels.

(Bregman projection). With a Euclidean map it reduces to projected gradient descent; with nonsmooth regularizers it yields proximal updates. This is particularly useful for pruning, where sparsity can be expressed as a regularizer (e.g., $\ell_1$, group, or block constraints) and enforced through a proximal step—stabilizing optimization while respecting structural constraints. Mirror descent also connects to continuous-time dynamics in differential-inclusion-based pruning, viewing proximal updates as discrete flows toward sparse sets. We build on this toolkit to unify local saliency with global sparsity budgets in an efficient, structure-aware pruning framework.

## 3 PROBLEM SETUP

**Overview of Post-Training Pruning**. Let $W_0$ be the pretrained parameters of an LLM and let $L(W)$ denote the task loss. Pruning seeks a sparse variable $\tilde{W} = M \odot W_0$ that preserves accuracy while reducing compute and memory, where $M \in \{0,1\}^d$ is a mask with target sparsity $s \in [0,1]$. We consider:

- Unstructured: elementwise masking with a global or per-layer budget.

- Semi-structured (N:M): in each contiguous M elements, at most N elements are kept.

A calibration set $\mathcal{C}$ is typically used to guide the choice of $M$, ensuring that the pruned model behaves similarly to the dense model at the desired sparsity and structure. Such calibration sets are often drawn from common pretraining corpora such as C4 (Raffel et al., 2020), WikiText (Merity et al., 2016), or PTB (Marcinkiewicz, 1994).

**Global feedback pruning.** Global feedback pruning directs sparsity decisions using the model's overall objective rather than isolated layer-wise signals. This avoids premature pruning, captures cross-layer dependencies, and produces sparser structures that better preserve performance. It uses a single, coordinated budget aligned with the model's global objective, guiding pruning toward a near-optimal solution rather than suboptimal local choices. However, it has limitations: mask-based formulations are often restricted to specific patterns (e.g., fixed ($N{:}M$)); optimization can be complex (involving auxiliary variables, alternating solvers, or regularized mask learning); and each run typically targets a single sparsity level, requiring separate executions for different budgets.

Let $\mathcal{D}$ measure the discrepancy between the pruned and dense models on $\mathcal{C}$, and let $Cost(M)$ be a global cost (e.g., nonzeros, FLOPs). A budgeted global objective is

$$\min_{M \in \mathcal{M}} \frac{1}{|\mathcal{C}|} \sum_{x \in \mathcal{C}} \mathcal{D}\big(f(x; W_0 \odot M),\ f(x; W_0)\big) \quad \text{s.t.} \quad Cost(M) \leq B, \tag{1}$$

---

**Algorithm 1** UniPruning: Mirror-Descent Pruning with Local Metric and Global Feedback

---

**Require:** Pretrained weights $W_0$; calibration set $\mathcal{C}$; parameters $\rho > 0$, $\kappa > 0$; total steps $N$; step sizes $\{\alpha_n\}_{n=0}^{N-1}$.

**Ensure:** Pruned weights $\widetilde{W}(B) = W_0 \odot \widehat{M}(B)$ for any global budget $B$.

1: **Local Saliency Statistics:** For each layer, run $\mathcal{C}$ once to collect inputs $X$. Compute local metric $S(W, X)$.

2: **Initialize:** $W^0 \leftarrow W_0$, $\Gamma^0 \leftarrow 0$, $V^0 \leftarrow 0$.

3: **for** $n = 0$ **to** $N - 1$ **do**

4:     $\textcolor{red}{S(W^n, X) \leftarrow \text{Local metric at current } W^n}$         $\textcolor{red}{\triangleright \text{ Recompute local statistics every iteration}}$

5:     $g_{\text{task}} \leftarrow \nabla_W \mathcal{L}_{\text{task}}(W^n)$

6:     $g_{\text{align}} \leftarrow \rho \cdot \nabla_W \frac{1}{2} \|\Gamma^n - S(W^n, X)\|_F^2$

7:     $W^{n+1} \leftarrow W^n - \kappa \alpha_n \cdot (g_{\text{task}} + g_{\text{align}})$         $\triangleright$ Gradient step on $W$

8:     **if** $N \!:\! M$ pruning **then**

9:         $W^{n+1} \leftarrow \text{Prox}_{R_{2:4}}(W^{n+1})$

10:         $R_{2:4}(w) = |w_1||w_2||w_3| + |w_2||w_3||w_4| + |w_3||w_4||w_1| + |w_4||w_1||w_2|$

11:     $V^{n+1} \leftarrow V^n - \alpha_n \rho \cdot (\Gamma^n - S(W^n, X))$

12:     $\Gamma^{n+1} \leftarrow \text{Prox}_\Omega(V^{n+1})$         $\triangleright$ Proximal update on $\Gamma$

13: $\Gamma^\star \leftarrow \Gamma^N$         $\triangleright$ Final saliency scores

**Export (unstructured):** Sort $|\Gamma^\star|$ once. For any global budget $B$, set threshold $\tau(B)$ to keep the top-$B$ entries and define mask $\widehat{M}(B) = \mathbb{I}(|\Gamma^\star| \geq \tau(B))$. Return $\widetilde{W}(B) = W_0 \odot \widehat{M}(B)$.

**Export ($N \!:\! M$):** In each block of size $M$, keep the top-$N$ entries by $|\Gamma^\star|$ and zero out the rest. Return $\widetilde{W}_{N:M} = W_0 \odot \widehat{M}_{N:M}$.

---

or, in Lagrangian form,

$$\min_{M \in \mathcal{M}} \frac{1}{|\mathcal{C}|} \sum_{x \in \mathcal{C}} \mathcal{D}\big(f(x; W_0 \odot M),\ f(x; W_0)\big)\ +\ \lambda\, \textcolor{red}{Cost(M)}. \tag{2}$$

Here, $\mathcal{M}$ encodes the structure (unstructured or $N \!:\! M$). This captures the "one budget for the whole model" view emphasized by global approaches.

**Local metric pruning.** Local metric pruning relies on simple heuristics, such as weight magnitude or weight activation products, to prune parameters independently within each layer. This makes it efficient and easy to apply without heavy global optimization or modifications to pretrained weights. It naturally supports multiple sparsity patterns (unstructured and semi-structured) and can operate in a single pass with only a small calibration set. However, this layer-wise independence comes at a cost: it ignores cross-layer dependencies and trade-offs, which can lead to suboptimal sparse structures and degraded model-level accuracy, particularly at high sparsity. Moreover, many criteria remain heuristic and lack a unified optimization framework, limiting principled control over sparsity allocation. So, its each layer $\ell$ selects a mask under its own budget $B_\ell$, $\textcolor{red}{\text{where } g_\ell \text{ represents the output of layer } \ell}$:

$$\min_{M_\ell \in \mathcal{M}_\ell} \frac{1}{|\mathcal{C}|} \sum_{x \in \mathcal{C}} \mathcal{D}_\ell\big(g_\ell(x; W_{0,\ell} \odot M_\ell),\ g_\ell(x; W_{0,\ell})\big) \quad \text{s.t.} \quad \textcolor{red}{Cost_\ell(M_\ell) \leq B_\ell}. \tag{3}$$

**UniPruning.** Our method offers the best of both worlds by unifying global feedback and local metric pruning within a mirror descent framework. It incorporates a lightweight, model-level controller that dynamically allocates pruning budgets across layers and enforces target sparsity patterns through a structured projection step. This design preserves the efficiency and flexibility of local pruning, leverages the coordination of global feedback, and remains easily tunable across diverse sparsity levels.

## 4 UNIPRUNING

We propose a mirror-descent pruning method that learns a saliency variable $\Gamma$ together with a trainable copy of the weights $W$ (initialized from $W_0$). After training, we sort the final $\Gamma$ once to derive

masks at any desired sparsity and apply those masks to $W_0$. In this way, $\Gamma$ acts as a data-driven pruning score. Because mask extraction is decoupled from training, the pretrained weights remain intact, which helps preserve performance without weight update.

## 4.1 ALGORITHM

**Local metric regularization**. We introduce a local metric regularizer that links the saliency score $\Gamma$ to the current weights $W$ by using a local importance metric $S(W, X)$, where $X$ denotes the input statistics collected from a small calibration set $\mathcal{C}$. This map encodes the local significance of each weight based on its interaction with the input activations, assigning higher scores to connections associated with stronger or more frequently activated inputs.

This score increases the importance of weights connected to strongly activated inputs, while reducing the influence of weights tied to weak or rarely used inputs. The absolute value ensures the score reflects the strength of the connection regardless of its sign, focusing only on the magnitude of the weight and the scale of the input activation.

This design follows the local metric approach used in Wanda (Sun et al., 2024), and can also incorporate other local methods such as RIA (Zhang et al., 2024); additional experimental comparisons are provided in the appendix.

To align the learned saliency $\Gamma$ with this local signal, we apply a simple alignment loss:

$$\left\| \Gamma - S(W, X) \right\|_F^2.$$

This encourages $\Gamma$ to reflect meaningful, data-driven local importance metric without constraining the weights themselves, which remain free to be updated by the task loss.

**Objective**. We now describe the training objective used throughout the pruning stage. Let the task loss on $\mathcal{C}$ be

$$\mathcal{L}_{\text{task}}(W) = \frac{1}{|\mathcal{C}|} \sum_{x \in \mathcal{C}} \ell\big(f(x; W)\big).$$

We consider the composite energy

$$\bar{\mathcal{L}}_\rho(W, \Gamma) = \mathcal{L}_{\text{task}}(W) + \tfrac{\rho}{2} \left\| \Gamma - S(W) \right\|_F^2 + \Omega(\Gamma), \tag{4}$$

where $\Omega$ is a sparsity-inducing term. The second term injects local metric by aligning $\Gamma$ with $S(W, X)$, $\rho$ as a hyperparameter; the third term imposes a global sparsity objective via $\Omega$, guiding the gradual gradient updates to $\Gamma$ and $W$ with global feedback. Because $\Omega$ can be non-differentiable such as $L_1$, we do not minimize equation 4 directly. Instead, we use a mirror-descent splitting that leads to the following dynamics.

**Dynamics and updates**. We now present the detailed algorithmic formulation of our proposed pruning method, UniPruning. During the sparsity training stage, we update the model weights $W$ and the saliency scores $\Gamma$ through a coupled dynamic process, with $V$ acts as a conjugate variable (or dual variable) . The weights $W$ follow a smooth gradient descent, while $\Gamma$ is updated via a proximal step guided by both local activation statistics and a global sparsity constraint.

$$W^{k+1} = W^k - \kappa \, \alpha_k \Big( \nabla_W \mathcal{L}_{\text{task}}(W^k) + \rho \, \nabla_W \tfrac{1}{2} \| \Gamma^k - S(W^k) \|_F^2 \Big), \tag{5}$$

$$V^{k+1} = V^k - \alpha_k \, \rho \big( \Gamma^k - S(W^k) \big), \tag{6}$$

$$\Gamma^{k+1} = \text{Prox}_\Omega \big( V^{k+1} \big). \tag{7}$$

Here the proximal operator of $\Omega$ is

$$\text{Prox}_\Omega(Z) = \arg\min_U \; \frac{1}{2} \| U - Z \|_F^2 + \Omega(U).$$

where $\rho$ and $\kappa$ are tunable hyperparameters, and $S(W)$ denotes a layer-wise saliency metric computed from activations. The procedure is initialized with $W^0 = W_0$, $\Gamma^0 = 0$, and $V^0 = 0$, and then proceeds iteratively as summarized in Algorithm 1.

The update on $\Gamma$ in equation 7 yields a saliency map, not a trajectory we retain. During training, the alignment term pulls $\Gamma_t$ toward the activation-aware scores $S(W_t)$, while the proximal map of $\Omega$

enforces sparsity and structure. As optimization proceeds, entries that matter for the task loss grow in magnitude in $\Gamma_t$, and unimportant ones are pushed toward zero. After the dynamics stabilizes, we treat the final map $\Gamma^\star$ as a data-driven ranking of connections. Sorting and thresholding $|\Gamma^\star|$ once produces masks at arbitrary sparsity levels, eliminating the need to retrain for each target.

After training converges, we discard all intermediate iterates and retain only the final $\Gamma^\star$. This mapping is then used to derive pruning masks at arbitrary sparsity levels without further retraining. Specifically, we sort $|\Gamma^\star|$ globally and select a threshold $\tau(B)$ that preserves the top-$B$ entries. The resulting mask and pruned weights are given by:

$$\widehat{M}(B)_{ij} = \mathbb{I}\left(|\Gamma^\star_{ij}| \geq \tau(B)\right), \quad \widetilde{W}(B) = W_0 \odot \widehat{M}(B).$$

Thus, a single training run suffices to generate pruning masks for any sparsity level, providing both flexibility and efficiency while avoiding repeated retraining.

## 4.2 DISCUSSION

Mirror descent stabilizes pruning by unifying local and global signals for robust high-sparsity performance. We give more insights here.

*(1) Why mirror descent is necessary.* Directly combining local metric and global feedback methods can introduce bias into the optimization process. To mitigate this, we decouple the model parameters from the sparsity objective, which stabilizes training and improves accuracy. This decoupling requires the introduction of an additional variable, $\Gamma$, to balance the trade-off between enforcing sparsity and preserving the convergence direction of the model. Ablation study of mirror descent is conducted in Section 5.3

*(2) Advantages of our method.* Our approach benefits from gradual, saliency-guided sparsification. By maintaining alignment with local metrics and leveraging the decoupled optimization framework, the model remains robust across architectures and pruning levels. As shown in our experiments 5, it maintains strong performance even at high sparsity ratios such as 60% and 70%, outperforming prior methods that often collapse under such conditions.

## 4.3 CONVERGENCE

Building on the mirror descent framework, we introduce a saliency variable $\Gamma$ as part of a splitting strategy to decouple weight optimization from sparsity enforcement. While this approach is empirically effective, it may influence the convergence behavior of mirror descent—particularly because the regularizer $\Omega$ is not necessarily differentiable. To address this, we provide a rigorous convergence analysis of our algorithm.

We study the convergence of the composite objective defined in Eq. 4, under the following assumptions: (i) $\mathcal{L}_{\text{task}}$ has Lipschitz continuous gradients and is bounded below, (ii) $S(W)$ is smooth with Lipschitz Jacobian, and (iii) $\Omega$ is proper, convex, and lower semi-continuous.

**Theorem 1** (Global convergence). *Under the above assumptions, if the step size $\alpha$ satisfies*

$$0 < \alpha < \tfrac{2}{\kappa(L_W + \rho L_S^2)},$$

*then the sequence $\{(W^k, \Gamma^k)\}$ generated by updates Eq. 7 converges to a critical point of Eq. 4.*

The proof of Theorem 1 is provided in Appendix A.5.1.

**Remark.** The convergence to a stationary point justifies extracting pruning masks directly from the limit $\Gamma^\star$ via global thresholding, enabling one-shot mask generation without retraining.

## 5 EXPERIMENTS

**Experimental Setup**. We evaluate on representative LLM families, including **LLaMA2** (Touvron et al., 2023), **Qwen2.5** (Team, 2025), and **Llama3** (Grattafiori et al., 2024) series, as well as distilled **DeepSeek** model (Guo et al., 2025). We consider several representative post-training pruning

Table 1: WikiText perplexity and zero-shot downstream benchmark results at 60% sparsity.

| Model | Method | WikiText PPL | ARC-C | ARC-E | HellaSwag | OBQA | PIQA | SIQA | Avg |
|---|---|---|---|---|---|---|---|---|---|
| | Dense | 4.57 | 0.4846 | 0.7942 | 0.6003 | 0.3500 | 0.7900 | 0.4729 | 0.5820 |
| | Magnitude | 11.22 | 0.2713 | 0.5623 | 0.4465 | 0.2180 | 0.6872 | 0.3941 | 0.4299 |
| LLaMA2-13B | Wanda | 11.90 | 0.3123 | 0.6460 | 0.4483 | 0.2740 | 0.7182 | 0.4243 | 0.4709 |
| | RIA | **7.57** | 0.3652 | 0.6970 | 0.5027 | **0.2840** | 0.7437 | 0.4524 | 0.5075 |
| | UniPruning | 7.82 | **0.3695** | **0.7003** | **0.5096** | **0.2840** | **0.7470** | **0.4529** | **0.5106** |
| | Dense | 6.39 | 0.4829 | 0.8047 | 0.6002 | 0.3360 | 0.7867 | 0.5481 | 0.5931 |
| | Magnitude | 3835.29 | 0.2295 | 0.3447 | 0.2594 | 0.1720 | 0.5305 | 0.3460 | 0.3137 |
| Qwen2.5-7B | Wanda | 14.06 | 0.3848 | 0.7163 | 0.4688 | **0.2680** | 0.7263 | **0.4780** | 0.5070 |
| | RIA | 12.09 | 0.3857 | **0.7344** | 0.4655 | 0.2600 | 0.7301 | 0.4621 | 0.5063 |
| | UniPruning | **11.87** | **0.3959** | 0.7306 | **0.4736** | 0.2620 | **0.7345** | 0.4703 | **0.5112** |
| | Dense | 4.93 | 0.5597 | 0.8241 | 0.6336 | 0.3480 | 0.8118 | 0.5537 | 0.6218 |
| | Magnitude | 117.74 | 0.3072 | 0.5455 | 0.4127 | 0.2900 | 0.6638 | 0.3828 | 0.4337 |
| Qwen2.5-14B | Wanda | 11.68 | 0.4266 | 0.7492 | **0.5070** | 0.3020 | 0.7595 | **0.4765** | 0.5368 |
| | RIA | 9.37 | 0.4360 | 0.7563 | 0.4991 | 0.2960 | **0.7601** | 0.4754 | 0.5378 |
| | UniPruning | **8.85** | **0.4531** | **0.7605** | **0.5070** | **0.3040** | 0.7601 | 0.4698 | **0.5424** |
| | Dense | 9.06 | 0.3157 | 0.6536 | 0.4774 | 0.2660 | 0.7459 | 0.4284 | 0.4812 |
| | Magnitude | 28096.59 | **0.1928** | 0.2643 | 0.2573 | 0.1380 | 0.5430 | 0.3373 | 0.2888 |
| Llama-3.2-1B | Wanda | 261.88 | 0.1852 | 0.2959 | 0.2701 | 0.1280 | 0.5555 | 0.3367 | 0.2952 |
| | RIA | 83.45 | **0.1928** | 0.3880 | 0.2884 | 0.1320 | 0.5941 | 0.3593 | 0.3258 |
| | UniPruning | **45.32** | 0.1886 | **0.4226** | **0.3104** | **0.1440** | **0.6153** | **0.3654** | **0.3411** |
| | Dense | 7.29 | 0.4224 | 0.7437 | 0.5532 | 0.3100 | 0.7666 | 0.4719 | 0.5446 |
| | Magnitude | 21913.05 | 0.2073 | 0.2639 | 0.2660 | 0.1380 | 0.5408 | 0.3296 | 0.2910 |
| Llama-3.2-3B | Wanda | 66.00 | 0.2193 | 0.4108 | 0.3132 | 0.1460 | 0.6121 | 0.3552 | 0.3428 |
| | RIA | 29.08 | 0.2483 | 0.5118 | 0.3555 | 0.1700 | 0.6659 | 0.3889 | 0.3901 |
| | UniPruning | **24.38** | **0.2654** | **0.5446** | **0.3714** | **0.1720** | **0.6774** | **0.4012** | **0.4053** |
| | Dense | 11.86 | 0.4087 | 0.7037 | 0.5554 | 0.3140 | 0.7606 | 0.4473 | 0.5316 |
| | Magnitude | 8154.51 | 0.2005 | 0.3066 | 0.2736 | 0.1520 | 0.5560 | 0.3444 | 0.3055 |
| DeepSeek-R1-Distill-Llama8B | Wanda | 50.66 | 0.2423 | 0.4676 | 0.3731 | 0.2000 | 0.6338 | 0.3864 | 0.3839 |
| | RIA | 28.16 | 0.2901 | 0.5358 | 0.4041 | 0.1960 | 0.6649 | 0.3987 | 0.4149 |
| | UniPruning | **24.50** | **0.3046** | **0.5804** | **0.4219** | **0.2140** | **0.6676** | **0.4186** | **0.4345** |

competitors which involve no weight update during pruning, keeping the same with our method: (1) Magnitude (Zhu & Gupta, 2017), the most prevalent pruning approach; (2) Wanda (Sun et al., 2024), which ranks weights using local metric scores and is applicable to both unstructured and semi-structured settings; (3) RIA (Zhang et al., 2024), which combines relative importance with activation norms to provide stable pruning decisions across different sparsity levels; and (4) ProxSparse (Liu et al., 2025), a proximal optimization framework that specifically targets semi-structured $N{:}M$ pruning and achieves state-of-the-art results under 2:4 pattern.

For fairness, we adopt common calibration setup of 128 randomly sampled C4 datasets (Raffel et al., 2020). Model quality is evaluated on both zero-shot reasoning tasks and language modeling: zero-shot performance is measured with the EleutherAI LM-Eval-Harnesss (Gao et al., 2024) on standard benchmark, while WikiText perplexity is reported as the language modeling metric (Merity et al., 2016). For unstructured pruning we use stochRIA (Yi & Richtárik, 2025) as the local metric, and for 2:4 semi-structured pruning we use Wanda (Sun et al., 2024). In both settings, we apply an additional $\lambda L_1$ regularization term $\Omega$ with $\lambda = 0.001$ (further discussed in Appendix A.4.3). Ablations of the local metric are provided in Section 5.3. For all LLMs, we fix the context length to 4096. All experiments are conducted on a single NVIDIA H200 GPU with 141GB of memory, using a learning rate of 1e-4.

## 5.1 UNSTRUCTURED PRUNING

We evaluate unstructured pruning at 60% sparsity on six pretrained LLMs, comparing UniPruning against Magnitude, Wanda, and RIA under the standard 128-sample C4 calibration. We report Wiki-Text perplexity and zero-shot accuracy on ARC-C/E, HellaSwag, OBQA, PIQA, and SIQA (Gao et al., 2024), and further include the average accuracy over all the evaluated benchmarks. Detailed results shows in Table 1.

Table 2: WikiText perplexity across models and pruning methods under 2:4 semi-structured pruning.

| Method | LLaMA2-13B | Qwen2.5-7B | Qwen2.5-14B | LLaMA-3.2-3B | DeepSeek-R1-Distill-Llama8B | DeepSeek-R1-Distill-Qwen-7B |
|---|---|---|---|---|---|---|
| Dense | 4.57 | 6.39 | 4.93 | 7.29 | 11.86 | 21.73 |
| Magnitude | 8.32 | inf | 48.59 | 668.75 | 459.82 | 270.25 |
| Wanda | 8.37 | 14.77 | 11.69 | 32.86 | 29.77 | 49.83 |
| RIA | 7.85 | 13.81 | 10.87 | 33.38 | 30.10 | 43.11 |
| ProxSparse | 6.88 | 14.06 | 10.54 | 22.44 | 23.74 | 35.66 |
| UniPruning | **6.87** | **10.86** | **9.10** | **21.20** | **20.91** | **30.24** |

Across six models, UniPruning attains the best average accuracy on every model, with only three per-task scores falling short of the top by small margins. For perplexity, UniPruning leads on 5 models, with the lone non-best case (LLaMA2-13B) trailing by just 0.25 (7.82 vs. 7.57).

In addition to average gains, Unified Pruning demonstrates stable improvements on commonsense-style tasks compared with other pruning baselines at the same sparsity level. For example, on Llama-3.2-3B, Unified Pruning improves ARC-C from 0.2193 (Wanda) and 0.2483 (RIA) to 0.2654, and SIQA from 0.3552 (Wanda) and 0.3512 (RIA) to 0.4012. On DeepSeek-R1-Distill-Llama8, it raises ARC-E from 0.4676 (Wanda) and 0.5358 (RIA) to 0.5804, and HellaSwag from 0.3731 (Wanda) and 0.4041 (RIA) to 0.4219. These results indicate that under high sparsity, **a globally coordinated budget allocation better preserves reasoning capacity**.

At 60% unstructured sparsity, UniPruning (i) sets **the best average zero-shot accuracy** on all reported bases with leading PPL and no collapse; and (ii) sustains **task-wise robustness** on commonsense benchmarks. These results support coupling local metric with a global budget to achieve balanced whole-model pruning.

## 5.2 SEMI-STRUCTURED $N:M$ PRUNING

For semi-structured pruning, we primarily evaluate the 2:4 sparsity pattern across models. To align with this hardware-friendly pattern, we incorporate a 2:4 regularizer (Kübler et al., 2025) into our algorithm. The detailed formulation and implementation of this adaptation are provided in Algorithm 1.

As shown in Table 2, Unified Pruning achieves the best perplexity across all evaluated models, consistently surpassing magnitude-based and importance-based baselines (e.g., Wanda, RIA). These results highlight the benefit of combining global coordination with local saliency in the semi-structured setting. Moreover, Unified Pruning also surpasses the current state-of-the-art ProxSparse, demonstrating that global coordination further enhances performance even under semi-structured constraints.

Beyond perplexity, we also evaluate zero-shot performance on downstream benchmarks under the 2:4 constraint, with results provided in Appendix A.2. Furthermore, to situate our approach against stronger baselines, we conduct an additional comparison with SparseGPT (Frantar & Alistarh, 2023), which allows weight updates during pruning. The corresponding results are also reported in Appendix A.2.

## 5.3 ABLATION STUDY

In this section, we conduct ablation studies to better understand the contribution of different components in our framework. We focus on two key aspects: (i) the choice of local saliency metric, which directly affects the pruning quality, and (ii) the role of mirror descent and the saliency variable, which are introduced to stabilize optimization and balance sparsity with task performance. These analyses provide deeper insights into the design choices underlying UniPruning and highlight their necessity.

**Local Metric.** To evaluate the sensitivity of pruning performance to the choice of local saliency metric, we conduct ablation studies comparing different saliency metric designs. Specifically, we experiment with magnitude-based scoring, activation-aware scoring as used in Wanda (Sun et al., 2024), and the combination of row/column norm-based relative importance with activation signals as

Table 3: WikiText perplexity of Qwen2.5-7B under different local metrics at varying sparsity.

| Local Metric | 50% | 60% | 70% |
|---|---|---|---|
| Magnitude | 38.86 | 428.56 | inf |
| Wanda | 8.63 | 13.12 | 183.21 |
| RIA | **8.28** | **11.87** | 88.35 |
| StochRIA | 8.63 | **11.87** | **52.34** |

Table 4: WikiText perplexity of Qwen2.5-7B in different $\Omega$ and $\rho$ at varying sparsity.

| Sparsity | UniPruning | $\lambda = 0.01, \rho = 10^{-5}$ | $\lambda = 0.01, \rho = 0$ | $\lambda = 0, \rho = 10^{-5}$ | $\lambda = 0, \rho = 0$ |
|---|---|---|---|---|---|
| 50% | 8.63 | 11.39 | 20.45 | 29.86 | 35.59 |
| 60% | 11.87 | 15.38 | 161.09 | inf | inf |

in RIA (Zhang et al., 2024). Furthermore, we incorporate the stochastic variant of RIA (stochRIA) proposed by (Yi & Richtárik, 2025), which introduces randomness into the scoring process to mitigate biases introduced by deterministic saliency measures and to enhance exploration during pruning.

Table 3 reports the results of different local saliency metrics on Qwen2.5-7B. Among the evaluated methods, stochRIA demonstrates a balanced trade-off across sparsity levels. At 50% and 60% sparsity, it performs comparably to RIA (8.63/11.87 vs. 8.28/11.87), and at 70% sparsity, it yields lower perplexity (52.34) than other alternatives. These results suggest that incorporating stochasticity into local importance estimation can improve robustness under high compression, making stochRIA a viable option for use within our framework.

**The Necessity of Mirror Descent.** As discussed in Section 4.2, we argue that mirror descent and the introduction of the saliency variable are necessary components of our framework. To verify this claim, we **additionally conduct pruning experiments using only the local metric and global feedback regularizers, without mirror descent or the saliency variable**. In other words, we directly train with the following objective function:

$$\bar{\mathcal{L}}_\rho(W) = \mathcal{L}_{\text{task}}(W) + \tfrac{\rho}{2} \| S(W) \|_F^2 + \Omega(W). \tag{8}$$

Since the $L_1$ regularizer is non-differentiable and cannot be directly integrated into the formulation without mirror descent, we replace $\Omega$ with a $\lambda L_2$ regularizer for this analysis. Table 4 reports WikiText perplexity results on Qwen2.5-7B under different $\Omega$ and $\rho$ configurations at sparsity levels of 50% and 60%.

As shown in Table 4, similarly, using stochRIA as the local saliency metric but removing the mirror descent update makes the optimization unstable and substantially increases perplexity, especially at higher sparsity levels. In the absence of local metric regularizer coefficient ($\rho = 0$) or $L_2$ coefficient ($\lambda = 0$), the model either diverges (infinite PPL) or converges to poor local minima. In contrast, our proposed UniPruning integrates local metric with a mirror descent driven global budget controller and achieves the lowest perplexity across all tested sparsity levels. These results demonstrate that both local saliency metrics and global coordination are essential. Crucially, mirror descent serves as the optimization bridge that enables their seamless integration in UniPruning, leading to stable and effective sparse pruning.

## 6 CONCLUSION

We presented UniPruning, a mirror–descent framework that unifies local evidence with global coordination to prune large language models. By introducing a saliency variable anchored to activation statistics and enforcing a model-wise sparsity budget, our method naturally supports both unstructured and semi-structured $N : M$ patterns, avoids direct weight updates, and enables one-shot mask extraction at multiple sparsity levels. Experiments show that Unified Pruning consistently delivers strong performance compared to prior baselines, while ablations highlight the necessity of mirror

descent and local saliency metric. Overall, the framework offers a principled, efficient, and scalable approach to LLM compression.

## ETHICS STATEMENT

This work does not involve human subjects, sensitive personal data, or applications with foreseeable societal risks. All experiments are conducted on publicly available datasets and widely used pretrained models under their respective licenses. We have carefully considered issues related to fairness, privacy, and potential misuse, and we believe our study poses minimal ethical concerns. According to ICLR policy, this section is excluded from the page limit.

## REPRODUCIBILITY STATEMENT

We have taken multiple steps to ensure reproducibility of our results. The paper clearly specifies the model architectures, training objectives, sparsity settings, and evaluation protocols. Additional implementation details, hyperparameters, and training scripts are provided in the supplementary materials. We also include pseudocode and references to relevant sections of the appendix to facilitate independent verification of our findings. According to ICLR policy, this section is excluded from the page limit.

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

# A    APPENDIX

## A.1    Use of Large Language Models

In preparing this paper, we leveraged GPT-4o to assist with polishing the writing, including grammar correction, stylistic consistency, and clarity of expression. All substantive elements—problem definition, methodological design, experimental execution, and analysis—were conceived and carried out solely by the authors. Every section of the manuscript was carefully reviewed and revised by the authors to guarantee fidelity to our original contributions. The authors take complete responsibility for the accuracy and integrity of the final text.

## A.2    Additional 2:4 pruning Results

As shown in Table 5, **Unified Pruning** consistently outperforms other pruning baselines across all evaluated models on downstream tasks. While naive magnitude pruning leads to severe degradation and Wanda or RIA offer only moderate improvements, Unified Pruning achieves the highest average accuracy among sparse methods and remains close to the dense baseline. For instance, on Qwen2.5-14B, Unified Pruning yields an average score of 0.5459, surpassing ProxSparse (0.5366). Similar trends hold for LLaMA-2 and DeepSeek variants, highlighting the robustness of our method across both medium- and large-scale LLMs.

Table 6 further evaluates language modeling perplexity under 2:4 semi-structured pruning. We observe that Unified Pruning outperforms SparseGPT on several benchmarks, such as Qwen2.5-7B (10.86 vs. 11.42) and DeepSeek-R1-Distill-LLaMA-8B (12.45 vs. 13.07), showing clear improvements. On other models the performance is slightly worse, e.g., LLaMA-3.2-3B (21.20 vs. 20.19), but the gap remains small and does not lead to collapse. Overall, Unified Pruning achieves competitive perplexity and can be comparable to, or even surpass, methods with weight update.

Taken together, these results demonstrate that coupling local saliency metrics with a unified global budget not only improves task-wise robustness under sparsity but also mitigates the perplexity blow-up commonly seen in post-training pruning methods. Unified Pruning thus offers a reliable path toward structured sparsification of LLMs while maintaining downstream task performance.

Figure 2: Wikitext perplexity comparison at 70% sparsity

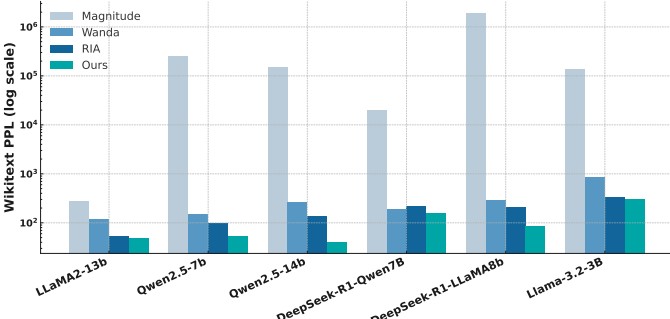

## A.3    Results On Ascend NPU

The open-source Pangu model based on Ascend has demonstrated powerful capabilities in various tasks. We conduct systematic empirical studies on the performance of the open-source Pangu model across multiple domains using the Ascend platform. Results are shown in Table 7. Our method outperforms others and interestingly finds the winning structure of the lottery ticket.

Table 5: Zero-shot downstream benchmark results under 2:4 pruning.

| Model | Method | ARC-C | ARC-E | HellaSwag | OBQA | PIQA | SIQA | Avg |
|---|---|---|---|---|---|---|---|---|
| LLaMA2-13B | Dense | 0.4846 | 0.7942 | 0.6003 | 0.3500 | 0.7900 | 0.4729 | 0.5820 |
| | Magnitude | 0.3174 | 0.6229 | 0.5011 | 0.2320 | 0.7171 | 0.4079 | 0.4664 |
| | Wanda | 0.3396 | 0.6856 | 0.4629 | 0.2460 | 0.7372 | 0.4243 | 0.4826 |
| | RIA | 0.3507 | 0.6987 | 0.4790 | 0.2600 | 0.7394 | 0.4294 | 0.4929 |
| | ProxSparse | 0.3695 | 0.6944 | **0.5300** | **0.2920** | 0.7427 | 0.4299 | 0.5098 |
| | UniPruning | **0.3712** | **0.7088** | 0.5255 | 0.2700 | **0.7535** | **0.4406** | **0.5116** |
| Qwen2.5-7B | Dense | 0.4829 | 0.8047 | 0.6002 | 0.3360 | 0.7867 | 0.5481 | 0.5931 |
| | Magnitude | 0.2432 | 0.3830 | 0.3006 | 0.2160 | 0.5560 | 0.3664 | 0.3442 |
| | Wanda | 0.3652 | 0.7142 | 0.4454 | 0.2600 | 0.7198 | **0.4703** | 0.4958 |
| | RIA | 0.3746 | 0.7176 | 0.4508 | 0.2680 | 0.7274 | 0.4678 | 0.5010 |
| | ProxSparse | **0.3985** | 0.7168 | **0.4803** | **0.2720** | 0.7296 | 0.4437 | 0.5068 |
| | UniPruning | 0.3959 | **0.7306** | 0.4736 | 0.2620 | **0.7345** | **0.4703** | **0.5112** |
| Qwen2.5-14B | Dense | 0.5597 | 0.8241 | 0.6336 | 0.3480 | 0.8118 | 0.5537 | 0.6218 |
| | Magnitude | 0.3584 | 0.6402 | 0.4176 | 0.2560 | 0.6790 | 0.4035 | 0.4591 |
| | Wanda | 0.3780 | 0.7231 | 0.4902 | 0.2840 | 0.7399 | 0.4386 | 0.5090 |
| | RIA | 0.3959 | 0.7386 | 0.4923 | 0.2720 | 0.7421 | 0.4545 | 0.5159 |
| | ProxSparse | 0.4428 | 0.7584 | 0.5208 | **0.2960** | 0.7470 | 0.4545 | 0.5366 |
| | UniPruning | **0.4531** | **0.7710** | **0.5268** | 0.2860 | **0.7617** | **0.4765** | **0.5459** |
| Llama-3.2-3B | Dense | 0.4224 | 0.7437 | 0.5532 | 0.3100 | 0.7666 | 0.4719 | 0.5446 |
| | Magnitude | 0.1954 | 0.3729 | 0.2837 | 0.1440 | 0.6055 | 0.3434 | 0.3242 |
| | Wanda | 0.2517 | 0.5093 | 0.3377 | 0.1640 | 0.6480 | 0.4675 | 0.3964 |
| | RIA | **0.2688** | 0.5311 | 0.3433 | 0.1760 | 0.6649 | 0.3823 | 0.3944 |
| | ProxSparse | 0.2611 | 0.5425 | **0.3877** | 0.1760 | **0.6768** | **0.3956** | 0.4066 |
| | UniPruning | 0.2602 | **0.5572** | 0.3797 | **0.1900** | 0.6746 | 0.3884 | **0.4084** |
| DeepSeek-R1-Distill-Llama8B | Dense | 0.4087 | 0.7037 | 0.5554 | 0.3140 | 0.7606 | 0.4473 | 0.5316 |
| | Magnitude | 0.2312 | 0.4242 | 0.3302 | 0.1340 | 0.6050 | 0.3495 | 0.3457 |
| | Wanda | 0.2756 | 0.5459 | 0.3852 | **0.1940** | 0.6534 | 0.4012 | 0.4092 |
| | RIA | 0.2816 | 0.5400 | 0.3846 | 0.1800 | 0.6659 | 0.3941 | 0.4077 |
| | ProxSparse | **0.2901** | 0.5366 | 0.4066 | 0.1760 | 0.6556 | 0.4002 | 0.4109 |
| | UniPruning | 0.2790 | **0.5614** | **0.4130** | 0.1880 | **0.6687** | **0.4069** | **0.4195** |
| DeepSeek-R1-Distill-Qwen-7B | Dense | 0.4232 | 0.6911 | 0.4637 | 0.26 | 0.7046 | 0.4248 | 0.4946 |
| | Magnitude | 0.2526 | 0.4848 | 0.3061 | 0.1640 | 0.5996 | 0.3562 | 0.3606 |
| | Wanda | 0.3046 | 0.5737 | 0.3660 | 0.1640 | 0.6491 | 0.3808 | 0.4064 |
| | RIA | 0.3166 | 0.5875 | 0.3686 | 0.1620 | 0.6583 | 0.3756 | 0.4114 |
| | ProxSparse | **0.3447** | 0.6330 | 0.3952 | **0.2120** | 0.6708 | 0.4023 | 0.4430 |
| | UniPruning | 0.3396 | **0.6545** | **0.4056** | 0.1940 | **0.6823** | 0.4023 | **0.4464** |

Table 6: Wikitext perplexity results across models and pruning methods under 2:4 pruning.

| Method | Weight Update | LLaMA2-13B | Qwen2.5-7B | Qwen2.5-14B | LLaMA-3.2-3B | DeepSeek-R1-Distill-Llama8B | DeepSeek-R1-Distill-Qwen-7B |
|---|---|---|---|---|---|---|---|
| Dense | - | 4.57 | 6.39 | 4.93 | 7.29 | 11.86 | 21.73 |
| SparseGPT | ✓ | 8.30 | **8.42** | 9.57 | **20.19** | 25.45 | 35.69 |
| UniPruning | ✗ | **6.87** | 10.86 | **9.10** | 21.20 | **20.91** | **30.24** |

## A.4 ADDITIONAL ANALYSIS

### A.4.1 THE ROBUST TO HIGHER SPARSITY.

We further evaluate pruning performance at a more aggressive **70% sparsity**, with results shown in Fig. 2. The gap between methods becomes more evident in this regime. Magnitude and Wanda both collapse under such high compression, leading to perplexities that grow by several orders of magnitude. RIA is more stable but still suffers from noticeable degradation. In contrast, UniPruning remains well-behaved across all tested architectures, consistently yielding perplexities within a reasonable range.

### A.4.2 INFERENCE EFFIENCY

We evaluated the throughput gain of applying 2:4 semi-structured sparsity to Qwen2.5-7B on an NVIDIA H200 GPU (batch size 8, sequence length 128). The sparse kernels accelerate the main compute-intensive modules: the self-attention projections (Q/K/V/O) achieve a **1.30×** speedup,

Table 7: Results On Ascend NPU.

| Model | Type | Method | ppl↓ | ARC-C↑ | ARC-E↑ | HellaSwag↑ | OBQA↑ | PIQA↑ | SIQA↑ | Avg↑ |
|---|---|---|---|---|---|---|---|---|---|---|
| openPangu-Embedded-7B-V1.1 | – | dense | 31.36 | 0.3302 | 0.5673 | 0.3946 | 0.4497 | 0.1980 | 0.6844 | 0.4374 |
| | 2:4 | ria | 208.04 | 0.2526 | 0.5034 | 0.3792 | 0.3645 | 0.1600 | 0.6344 | 0.3824 |
| | 2:4 | wanda | 237.32 | 0.2628 | 0.5109 | 0.3746 | 0.3593 | 0.1600 | 0.6328 | 0.3834 |
| | 2:4 | UniPruning | **106.21** | **0.2927** | **0.6002** | **0.3930** | **0.3778** | **0.1840** | **0.6518** | **0.4166** |
| | 50% | ria | 59.75 | 0.3072 | 0.5488 | 0.3936 | 0.4211 | 0.1960 | 0.6632 | 0.4217 |
| | 50% | wanda | 70.26 | 0.2944 | 0.5492 | 0.3930 | 0.4227 | **0.2060** | 0.6670 | 0.4221 |
| | 50% | UniPruning | **49.73** | **0.3677** | **0.7054** | **0.4043** | **0.6700** | 0.2040 | **0.6959** | **0.5079** |

while the MLP blocks (up, down, and gating projections) reach **1.34×**. When integrating all components—including non-sparse operations such as softmax, normalization, and key–value I/O, the overall end-to-end inference achieves a **1.27×** throughput improvement. These results fall within the typical **1.2–1.4×** range reported for 2:4 sparsity on modern accelerators.

Table 8: Inference time analysis for Qwen2.5-7B.

| Module name | Speedup ratio |
|---|---|
| self_attn Q/K/V/O | 1.30× |
| MLP up/down/gate | 1.34× |
| End-to-end inference | **1.27×** |

### A.4.3 HYPERPARAMETER ANALYSIS

Figure 3: WikiText perplexity of different models at 60% sparsity across $\lambda$ values.

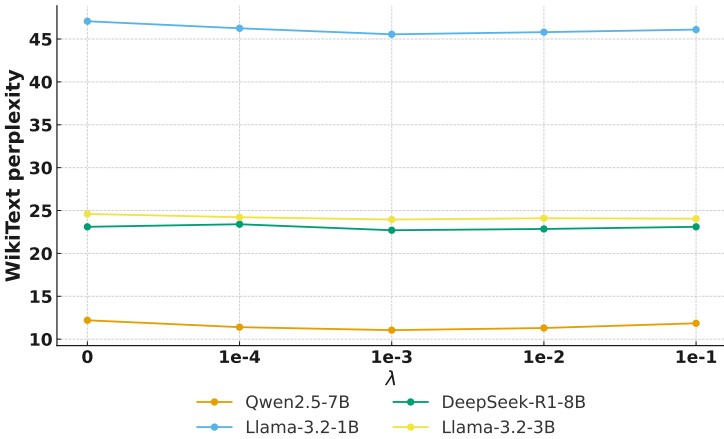

We further investigate the impact of key hyperparameters on the performance of our method. In particular, we examine the effects of the regularization weight $\lambda$. Results shown in Fig. 3. These experiments provide insights into the sensitivity of UniPruning to hyperparameter choices and demonstrate the robustness of our framework under different configurations.

### A.4.4 LIMITATIONS OF OUR METHOD

While UniPruning achieves strong performance across various model families and sparsity regimes, several limitations remain that warrant further investigation:

- **Additional hyperparameters.** Our framework introduces additional hyperparameters, including the regularization coefficient $\lambda$ and the choice of local saliency metric. As demonstrated in our hyperparameter analysis, model performance can vary across different configurations.
- **Limited architectural generalization.** Our experiments primarily focus on LLaMA and Qwen model families, with limited exploration of other transformer architectures. It remains an open question how well the proposed method generalizes to models with substantially different design paradigms.

These limitations highlight promising directions for improving the robustness and applicability of Unified Pruning in broader model and deployment contexts.

## A.5 PROOF OF THEOREM 1

First of all, we reformulate Eq. 4 into an equivalent form. Without loss of generality, consider $\Omega = L_1$ in the sequel. Denote $R(P) := \Omega(\Gamma)$, then our UniPruning Algorithm is equivalent to the following iterations,

$$W_{k+1} = W_k - \kappa\alpha\nabla_W\bar{\mathcal{L}}(W_k, \Gamma_k), \tag{9a}$$

$$\Gamma_{k+1} = \text{Prox}_{\kappa\Omega}(\Gamma_k + \kappa(g_k - \alpha\nabla_\Gamma\bar{\mathcal{L}}(W_k, \Gamma_k))), \tag{9b}$$

$$g_{k+1} = g_k - \kappa^{-1}(\Gamma_{k+1} - \Gamma_k + \kappa\alpha \cdot \nabla_\Gamma\bar{\mathcal{L}}(W_k, \Gamma_k)). \tag{9c}$$

where $p_k = [0, g_k]^T \in \partial R(P_k)$ and $g_k \in \partial\Omega(\Gamma_k)$. Thus

The global convergence of $(M_k, \Gamma_k, g_k)$ can be established based on the Kurdyka-Łojasiewicz framework.

### A.5.1 SUFFICIENT DESCENT PROPERTY ALONG LYAPUNOV FUNCTION

Let $P_k := (M_k, \Gamma_k)$, and $Q_k := (P_k, g_{k-1}), k \in \mathbb{N}$. In the following, we present the sufficient descent property of $Q_k$ along the Lyapunov function $F$.

**Lemma.** Suppose that $\mathcal{L}$ is continuously differentiable and $\nabla\mathcal{L}$ is Lipschitz continuous with a constant $Lip > 0$, $C = \max|W_0|$ is the max value of the pretrained model parameters $W_0$. Let $\{Q_k\}$ be a sequence generated by SLBI with a finite initialization. If $0 < \alpha < \frac{2}{\kappa(Lip*C+\nu^{-1})}$, then

$$F(Q_{k+1}) \leq F(Q_k) - \rho\|Q_{k+1} - Q_k\|_2^2,$$

where $\rho := \frac{1}{\kappa} - \frac{\alpha(Lip*C+\nu^{-1})}{2}$.

*Proof.* By the optimality condition of equation 7 and also the inclusion $p_k = [0, g_k]^T \in \partial R(P_k)$, there holds

$$\kappa(\alpha\nabla\bar{\mathcal{L}}(P_k) + p_{k+1} - p_k) + P_{k+1} - P_k = 0,$$

which implies

$$\nabla\hat{\mathcal{L}}(M) = \sum\nabla\mathcal{L}(\hat{W}) * W_0 \tag{10}$$

Noting that $\bar{\mathcal{L}}(P) = \hat{\mathcal{L}}(M) + \frac{1}{2\nu}\|M - \Gamma\|_2^2 = \mathcal{L}(W_0 \odot M) + \frac{1}{2\nu}\|M - \Gamma\|_2^2$. Together with,

$$\alpha\bar{\mathcal{L}}(P_{k+1}) + D(\Gamma_{k+1}, \Gamma_k) + \rho\|P_{k+1} - P_k\|_2^2 \leq \alpha\bar{\mathcal{L}}(P_k). \tag{11}$$

Adding some terms in both sides of the above inequality and after some reformulations implies

$$F(Q_{k+1}) \leq F(Q_k) - \rho\|P_{k+1} - P_k\|_2^2 - B_\Omega^{g_{k+1}}(\Gamma_k, \Gamma_{k-1}) - B_\Omega^{g_{k-1}}(\Gamma_k, \Gamma_{k-1}) \tag{12}$$

$$\leq F(Q_k) - \rho\|P_{k+1} - P_k\|_2^2, \tag{13}$$

where the final equality holds for $D(\Gamma_{k+1}, \Gamma_k) - B_\Omega^{g_k}(\Gamma_{k+1}, \Gamma_k) = B_\Omega^{g_{k+1}}(\Gamma_k, \Gamma_{k-1})$.

Note that the final inequality holds for $B_\Omega^{g_{k+1}}(\Gamma_k, \Gamma_{k-1}) \geq 0$ and $B_\Omega^{g_{k-1}}(\Gamma_k, \Gamma_{k-1}) \geq 0$. Thus, we finish the proof of this lemma. □

Based on Lemma A.5.1, we directly obtain the following lemma.

**Lemma 2.** *Suppose that assumptions of Lemma A.5.1 hold. Then*

> *(i) both $\alpha\{\bar{\mathcal{L}}(P_k)\}$ and $\{F(Q_k)\}$ converge to the same finite value, and $\lim_{k\to\infty} B_\Omega^{g_k}(\Gamma_{k+1}, \Gamma_k) = 0$.*

*(ii) the sequence $\{(M_k, \Gamma_k, g_k)\}$ is bounded,*

*(iii)* $\lim_{k\to\infty} \|P_{k+1} - P_k\|_2^2 = 0$ *and* $\lim_{k\to\infty} D(\Gamma_{k+1}, \Gamma_k) = 0$,

*(iv)* $\frac{1}{K} \sum_{k=0}^{K} \|P_{k+1} - P_k\|_2^2 \to 0$ *at a rate of* $\mathcal{O}(1/K)$.

*Proof.* By (11), $\bar{\mathcal{L}}(P_k)$ is monotonically decreasing due to $D(\Gamma_{k+1}, \Gamma_k) \geq 0$. Similarly, by (13), $F(Q^k)$ is also monotonically decreasing. By the lower boundedness assumption of $\mathcal{L}(W)$, both $\bar{\mathcal{L}}(P)$ and $F(Q)$ are lower bounded by their definitions respectively. Therefore, both $\{\bar{\mathcal{L}}(P_k)\}$ and $\{F(Q_k)\}$ converge, and it is obvious that $\lim_{k\to\infty} F(Q_k) \geq \lim_{k\to\infty} \alpha\bar{\mathcal{L}}(P_k)$. By (12),

$$B_\Omega^{g_{k-1}}(\Gamma_k, \Gamma_{k-1}) \leq F(Q_k) - F(Q_{k+1}), \ k = 1, \ldots.$$

By the definition of $F(Q_k) = \alpha\bar{\mathcal{L}}(P_k) + B_\Omega^{g_{k-1}}(\Gamma_k, \Gamma_{k-1})$ and the above equality, it yields

$$\lim_{k\to\infty} F(Q_k) = \lim_{k\to\infty} \alpha\bar{\mathcal{L}}(P_k).$$

Since $L(M)$ has bounded level sets, then $M_k$ is bounded. By the definition of $\bar{\mathcal{L}}(M, \Gamma)$ and the finiteness of $\bar{\mathcal{L}}(M_k, \Gamma_k)$, $\Gamma_k$ is also bounded due to $M_k$ is bounded. The boundedness of $g_k$ is due to $g_k \in \partial\Omega(\Gamma_k)$, condition (d), and the boundedness of $\Gamma_k$.

By (13), summing up (13) over $k = 0, 1, \ldots, K$ yields

$$\sum_{k=0}^{K} \left(\rho\|P_{k+1} - P_k\|^2 + D(\Gamma_{k+1}, \Gamma_k)\right) < \alpha\bar{\mathcal{L}}(P_0) < \infty. \tag{14}$$

Letting $K \to \infty$ and noting that both $\|P_{k+1} - P_k\|^2$ and $D(\Gamma_{k+1}, \Gamma_k)$ are nonnegative, thus

$$\lim_{k\to\infty} \|P_{k+1} - P_k\|^2 = 0, \quad \lim_{k\to\infty} D(\Gamma_{k+1}, \Gamma_k) = 0.$$

Again by (14),

$$\frac{1}{K} \sum_{k=0}^{K} \left(\rho\|P_{k+1} - P_k\|^2 + D(\Gamma_{k+1}, \Gamma_k)\right) < K^{-1}\alpha\bar{\mathcal{L}}(P_0),$$

which implies $\frac{1}{K} \sum_{k=0}^{K} \|P_{k+1} - P_k\|^2 \to 0$ at a rate of $\mathcal{O}(1/K)$. $\qquad\square$

### A.5.2 RELATIVE ERROR PROPERTY

In this subsubsection, we provide the bound of subgradient by the discrepancy of two successive iterates.

$$H_{k+1} := \begin{pmatrix} \alpha\nabla_M\bar{\mathcal{L}}(M_{k+1}, \Gamma_{k+1}) \\ \alpha\nabla_\Gamma\bar{\mathcal{L}}(M_{k+1}, \Gamma_{k+1}) + g_{k+1} - g_k \\ \Gamma_k - \Gamma_{k+1} \end{pmatrix} \in \partial F(Q_{k+1}), \ k \in \mathbb{N}. \tag{15}$$

**Lemma.** Under assumptions of Lemma 2, then

$$\|H_{k+1}\| \leq \rho_1\|Q_{k+1} - Q_k\|, \text{ for } H_{k+1} \in \partial F(Q_{k+1}), \ k \in \mathbb{N},$$

where $\rho_1 := 2\kappa^{-1} + 1 + \alpha(Lip * C + 2\nu^{-1})$. Moreover, $\frac{1}{K} \sum_{k=1}^{K} \|H_k\|^2 \to 0$ at a rate of $\mathcal{O}(1/K)$.

*Proof.* Note that

$$\nabla_M\bar{\mathcal{L}}(M_{k+1}, \Gamma_{k+1}) = (\nabla_M\bar{\mathcal{L}}(M_{k+1}, \Gamma_{k+1}) - \nabla_M\bar{\mathcal{L}}(M_{k+1}, \Gamma_k)) \tag{16}$$
$$+ (\nabla_M\bar{\mathcal{L}}(M_{k+1}, \Gamma_k) - \nabla_M\bar{\mathcal{L}}(M_k, \Gamma_k)) + \nabla_M\bar{\mathcal{L}}(M_k, \Gamma_k).$$

where the last inequality holds for the Lipschitz continuity of $\nabla\mathcal{L}$ with a constant $Lip > 0$, and $C = \max|W_0|$. By equation 9a,

$$\|\nabla_M\bar{\mathcal{L}}(M_k, \Gamma_k)\| = (\kappa\alpha)^{-1}\|M_{k+1} - M_k\|.$$

Substituting the above (in)equalities into (16) yields

$$\|\nabla_M \bar{\mathcal{L}}(M_{k+1}, \Gamma_{k+1})\| \leq \left[(\kappa\alpha)^{-1} + Lip * C + \nu^{-1}\right] \cdot \|M_{k+1} - M_k\| + \nu^{-1}\|\Gamma_{k+1} - \Gamma_k\|$$

Thus,

$$\|\alpha\nabla_M \bar{\mathcal{L}}(M_{k+1}, \Gamma_{k+1})\| \leq \left[\kappa^{-1} + \alpha(Lip * C + \nu^{-1})\right] \cdot \|M_{k+1} - M_k\| + \alpha\nu^{-1}\|\Gamma_{k+1} - \Gamma_k\|. \tag{17}$$

Noting that $\nabla_\Gamma \bar{\mathcal{L}}(M_k, \Gamma_k) = \nu^{-1}(\Gamma_k - M_k)$, and after some simplifications yields

$$\|\alpha\nabla_\Gamma \bar{\mathcal{L}}(M_{k+1}, \Gamma_{k+1}) + g_{k+1} - g_k\| = \|(\kappa^{-1} - \alpha\nu^{-1}) \cdot (\Gamma_k - \Gamma_{k+1}) + \alpha\nu^{-1}(M_k - M_{k+1})\|$$
$$\leq \alpha\nu^{-1}\|M_k - M_{k+1}\| + (\kappa^{-1} - \alpha\nu^{-1})\|\Gamma_k - \Gamma_{k+1}\|, \tag{18}$$

where the last inequality holds for the triangle inequality and $\kappa^{-1} > \alpha\nu^{-1}$ by the assumption.

By (17), (18), and the definition of $H_{k+1}$ (15), there holds

$$\|H_{k+1}\| \leq \left[\kappa^{-1} + \alpha(Lip * C + 2\nu^{-1})\right] \cdot \|M_{k+1} - M_k\| + (\kappa^{-1} + 1)\|\Gamma_{k+1} - \Gamma_k\|$$
$$\leq \left[2\kappa^{-1} + 1 + \alpha(Lip * C + 2\nu^{-1})\right] \cdot \|Q_{k+1} - Q_k\|.$$

By Lemma 2(iv), $\frac{1}{K}\sum_{k=1}^K \|H_k\|^2 \to 0$ at a rate of $\mathcal{O}(1/K)$.

This finishes the proof of this lemma. $\qquad\square$

### A.5.3 KURDYKA-ŁOJASIEWICZ PROPERTY

To introduce the definition of the Kurdyka-Łojasiewicz (KL) property, we need some notions and notations from variational analysis.

The notion of subdifferential plays a central role in the following definitions. For each $\mathbf{x} \in \text{dom}(h) := \{\mathbf{x} \in \mathbb{R}^p : h(\mathbf{x}) < +\infty\}$, the *Fréchet subdifferential* of $h$ at $\mathbf{x}$, written $\widehat{\partial}h(\mathbf{x})$, is the set of vectors $\mathbf{v} \in \mathbb{R}^p$ which satisfy

$$\lim_{\mathbf{y} \neq \mathbf{x}, \mathbf{y} \to \mathbf{x}} \inf \frac{h(\mathbf{y}) - h(\mathbf{x}) - \langle \mathbf{v}, \mathbf{y} - \mathbf{x}\rangle}{\|\mathbf{x} - \mathbf{y}\|} \geq 0.$$

When $\mathbf{x} \notin \text{dom}(h)$, we set $\widehat{\partial}h(\mathbf{x}) = \varnothing$. The *limiting-subdifferential* (or simply *subdifferential*) of $h$, written $\partial h(\mathbf{x})$ at $\mathbf{x} \in \text{dom}(h)$, is defined by

$$\partial h(\mathbf{x}) := \{\mathbf{v} \in \mathbb{R}^p : \exists \mathbf{x}^k \to \mathbf{x}, \ h(\mathbf{x}^k) \to h(\mathbf{x}), \ \mathbf{v}^k \in \widehat{\partial}h(\mathbf{x}^k) \to \mathbf{v}\}. \tag{19}$$

A necessary (but not sufficient) condition for $\mathbf{x} \in \mathbb{R}^p$ to be a minimizer of $h$ is $\mathbf{0} \in \partial h(\mathbf{x})$. A point that satisfies this inclusion is called *limiting-critical* or simply *critical*. The distance between a point $\mathbf{x}$ to a subset $\mathcal{S}$ of $\mathbb{R}^p$, written $\text{dist}(\mathbf{x}, \mathcal{S})$, is defined by $\text{dist}(\mathbf{x}, \mathcal{S}) = \inf\{\|\mathbf{x} - \mathbf{s}\| : \mathbf{s} \in \mathcal{S}\}$, where $\|\cdot\|$ represents the Euclidean norm.

The *graph* is defined by

$$\text{Graph}(h) := \{(\mathbf{x}, y) \in \mathbb{R}^p \times \mathbb{R} : y = h(\mathbf{x})\},$$
$$(\text{resp. Graph}(h) := \{(\mathbf{x}, \mathbf{y}) \in \mathbb{R}^p \times \mathbb{R}^q : \mathbf{y} \in h(\mathbf{x})\}),$$

and its domain by $\text{dom}(h) := \{\mathbf{x} \in \mathbb{R}^p : h(\mathbf{x}) < +\infty\}$ (resp. $\text{dom}(h) := \{\mathbf{x} \in \mathbb{R}^p : h(\mathbf{x}) \neq \varnothing\}$). When $h$ is a proper function, i.e., when $\text{dom}(h) \neq \varnothing$, the set of its global minimizers (possibly empty) is denoted by

$$\arg\min h := \{\mathbf{x} \in \mathbb{R}^p : h(\mathbf{x}) = \inf h\}.$$

**Definition 3.** *[Kurdyka-Łojasiewicz property] A function $h$ is said to have the Kurdyka-Łojasiewicz (KL) property at $\bar{u} \in \text{dom}(\partial h) := \{v \in \mathbb{R}^n | \partial h(v) \neq \emptyset\}$, if there exists a constant $\eta \in (0, \infty)$, a neighborhood $\mathcal{N}$ of $\bar{u}$ and a function $\phi : [0, \eta) \to \mathbb{R}_+$, which is a concave function that is continuous at $0$ and satisfies $\phi(0) = 0$, $\phi \in \mathcal{C}^1((0, \eta))$, i.e., $\phi$ is continuous differentiable on $(0, \eta)$, and $\phi'(s) > 0$ for all $s \in (0, \eta)$, such that for all $u \in \mathcal{N} \cap \{u \in \mathbb{R}^n | h(\bar{u}) < h(u) < h(\bar{u}) + \eta\}$, the following inequality holds*

$$\phi'(h(u) - h(\bar{u})) \cdot \text{dist}(0, \partial h(u)) \geq 1. \tag{20}$$

*If $h$ satisfies the KL property at each point of $\text{dom}(\partial h)$, $h$ is called a KL function.*

**Definition 4.** *[Semialgebraic]*

(a) *A function $h : \mathbb{R}^p \to \mathbb{R} \cup \{+\infty\}$ (resp. a point-to-set mapping $h : \mathbb{R}^p \rightrightarrows \mathbb{R}^q$) is called semialgebraic if its graph $\mathrm{Graph}(h)$ is a semialgebraic set.*

(b) *A set $\mathcal{D} \subset \mathbb{R}^p$ is called semialgebraic if it can be represented as*

$$\mathcal{D} = \bigcup_{i=1}^{s} \bigcap_{j=1}^{t} \{\mathbf{x} \in \mathbb{R}^p : P_{ij}(\mathbf{x}) = 0, Q_{ij}(\mathbf{x}) > 0\},$$

*where $P_{ij}, Q_{ij}$ are real polynomial functions for $1 \le i \le s, 1 \le j \le t$.*

The class of semialgebraic sets are stable under the operation of finite union, finite intersection, Cartesian product or complementation. Some typical examples include polynomial functions, the indicator function of a semialgebraic set, and the Euclidean norm.

**Definition 5.** *[Real analytic] A function $h$ with domain an open set $U \subset \mathbb{R}$ and range the set of either all real or complex numbers, is said to be **real analytic** at $u$ if the function $h$ may be represented by a convergent power series on some interval of positive radius centered at $u$: $h(x) = \sum_{j=0}^{\infty} \alpha_j (x - u)^j$, for some $\{\alpha_j\} \subset \mathbb{R}$. The function is said to be **real analytic** on $V \subset U$ if it is real analytic at each $u \in V$. The real analytic function $f$ over $\mathbb{R}^p$ for some positive integer $p > 1$ can be defined similarly.*

*Typical real analytic functions include polynomials, exponential functions, and the logarithm, trigonometric and power functions on any open set of their domains. One can verify whether a multivariable real function $h(\mathbf{x})$ on $\mathbb{R}^p$ is analytic by checking the analyticity of $g(t) := h(\mathbf{x} + t\mathbf{y})$ for any $\mathbf{x}, \mathbf{y} \in \mathbb{R}^p$.*

Let $(\bar{W}, \bar{\Gamma}, \bar{g})$ be a critical point of $F$, then the following holds

$$\begin{aligned}
\partial_M F(\bar{M}, \bar{\Gamma}, \bar{g}) &= \alpha(\nabla \mathcal{L}(\bar{M}) + \nu^{-1}(\bar{M} - \bar{\Gamma})) = 0, \\
\partial_\Gamma F(\bar{M}, \bar{\Gamma}, \bar{g}) &= \alpha \nu^{-1}(\bar{\Gamma} - \bar{M}) + \partial \Omega(\bar{\Gamma}) - \bar{g} \ni 0, \\
\partial_g F(\bar{M}, \bar{\Gamma}, \bar{g}) &= \bar{\Gamma} - \partial \Omega^*(\bar{g}) \ni 0.
\end{aligned} \tag{21}$$

By the final inclusion and the convexity of $\Omega$, it implies $\bar{g} \in \partial \Omega(\bar{\Gamma})$. Plugging this inclusion into the second inclusion yields $\alpha \nu^{-1}(\bar{\Gamma} - \bar{M}) = 0$. Together with the first equality imples

$$\nabla \bar{\mathcal{L}}(\bar{M}, \bar{\Gamma}) = 0, \quad \nabla \mathcal{L}(\bar{M}) = 0.$$

This finishes the proof of this theorem.

