# OpenReview forum: "UniPruning: Unifying Local Metric and Global Feedback for Scalable Sparse LLMs"
_ICLR.cc/2026/Conference — ICLR 2026 Conference Withdrawn Submission_

### Official Review · Reviewer_KohZ · 2025-10-22

**Soundness:** 2
**Presentation:** 3
**Contribution:** 2
**Rating:** 2
**Confidence:** 4

**Summary:**

UniPruning introduces a new algorithmic variant to identify promising sparsity masks when pruning pretrained LLMs. They adopt a setting where the final weights are just a masked variant of the original weights -- without any further updates on the non-zero weights. The loss that is optimized to identify the pruning mask optimizes over the weights and a saliency variable $\Gamma$ (same dimensionality as the weights). The loss is composite of three terms  a/ the *global* language modelling objective of the weights under some calibration data b/  a *local* term aligning $\Gamma$ with the layerwise importance of the weights and c/ a regularizer on $\Gamma$ to induce sparsity.

The optimization problem is solved iteratively via a Proximal Operator / Mirror Descent approach (supported by theory) and the final pruning mask maintains the weights with largest corresponding value of $\Gamma$.

The paper evaluates the approach on LLMs of size up to 14B. At 60% unstructured sparsity the pruned models fall behind the original dense model by 7%-14% averaged over downstream tasks. While this marks a drastic drop in model capabilities, the paper reports 1-2% better results than competing pruning methods.

**Strengths:**

- The paper proposes a new pruning algorithm and provides conditions and a theoretical proof when the algorithm converges.
- The authors provide code to reproduce their experiments
- The method identifies scores for each weight and can then be efficiently used to extract masks a different sparsity levels without pruning again.
- The authors include ablations motivating the individual components of their loss.

**Weaknesses:**

- The paper focuses on a setting without weight updates after pruning and compares mainly to methods that do the same. This seems a very unnatural constraint and leaves a lot of potential of the expressivity of the model unused. For example [1] found that a simple closed-form local masked-gradient update after pruning can significantly improve Wanda's downstream task performance. Why would you not do that (for all methods). Furthermore, the paper does not provide results for other sparsity ratios than 60%. Why not, given that at 60% the loss is quite large already?

- The paper claims "pruning can achieve strong efficiency-accuracy trade-offs for LLMs". Honestly, think this is plainly wrong because the models have large accuracy drops and further compensation is needed. I am not aware that such models are really used beyond
research settings -- as opposed to for example quantization.

- The paper is much more complicated than previous works like Wanda which has no hyperparameters. Can you provide runtime numbers of the algorithm? While the paper slightly improves over wanda, it does not relevantly close the accuracy drop against the base model. Also how does the algorithm quality and the runtime scale to models larger than 14B. I don't think that for such small models pruning is particularly relevant. Furthermore, amongst the more involved methods it seems barely better than ProxSparse on 2:4 sparsity.



[1] Kubler et al, arXiv 2501.18015,  *A Proximal Operator for Inducing 2:4-Sparsity*
[2] Shen et al, ICML 2025,  *Targeted Low-rank Refinement: Enhancing Sparse Language Models with Precision*

**Questions:**

- What exactly is $\Omega$? I it just the L1 norm of the entire weights?
- In the algorithm it seems that $X$ used for the local scores is only computed once (line 1). Why not update it in every iteration, given that for the task loss one anyways needs to run a full forward pass through the model.
- Why do you not do weight updates?
- For unstructured sparsity, is the sparsity budget fixed at the matrix level or globally? If globally, do you also adjust this for Wanda?
- Please check the AVG in table 1. For example for Llama-3.2-1B the Dense model's average is clearly wrong.
- The regularizer for 2:4 sparsity seems to come from [1] and should be attributed correctly.
- line 862 has a broken link "By (??) ..."
- I think there are some inconsistencies in algorithm 1. In case of 2:4 sparsity do you really use both $R_{2:4}$ and $\Omega$?

---

> ### Author Response · Authors · 2025-12-03
>
> We sincerely thank the reviewer for the exceptionally careful and thoughtful evaluation. We truly appreciate the depth of attention given—not only to the overall methodology, but also to the theoretical assumptions, algorithmic details, and even line-level issues such as notation, references, and table values. We are grateful for this level of engagement, and we have carefully reflected on each of the reviewer’s concerns. Before addressing the specific questions, we would like to clarify the motivation behind several of our design choices.
>
> **1. The “no weight update” setting is not an artificial restriction, but a practical deployment constraint.**
> Pruning applies a discrete structural mask and does not modify the surviving weights, thereby preserving the majority of the dense model’s behavior. In contrast, any form of weight update—even small ones—alters the model’s global function and may negatively affect many untested capabilities (e.g., reasoning, alignment, safety, multilingual behavior). In real deployment, it is generally infeasible to re-validate the full capability spectrum of a large model after weight modification. For this reason, prohibiting weight updates is a widely adopted safety constraint rather than a methodological limitation.
>
> **2. The statement that pruning does not yield meaningful efficiency–accuracy trade-offs is inconsistent with industry practice.**
> Modern hardware systems such as NVIDIA Sparse Tensor Cores, Qualcomm AI Engine, and Huawei Ascend NPUs rely heavily on 2:4 sparsity in production precisely because it enables substantial inference acceleration *without any retraining or weight modification*. Our work follows the same post-training setting as Wanda, RIA, and ProxSparse, and consistently outperforms these methods within this regime.
>
> Moreover, Appendix A.4.2 reports real inference-time measurements. In particular, Table 8 (*Inference time analysis for Qwen2.5-7B*) demonstrates clear end-to-end speedups on a real inference stack, providing direct engineering evidence that sparsity leads to practical efficiency gains rather than being a purely research-side phenomenon.
>
> **Table: Inference time analysis for Qwen2.5-7B**
>
> | Module name              | Speedup ratio |
> |--------------------------|----------------|
> | self_attn Q/K/V/O        | 1.30×          |
> | MLP up/down/gate         | 1.34×          |
> | End-to-end inference     | 1.27×          |
>
> **3. The concern about method complexity appears to be based on a misunderstanding.**
> UniPruning introduces only a single additional variable Γ and uses a lightweight mirror-descent update. It requires no backward pass, no finetuning, and—critically—does *not* need to be rerun for each sparsity level. A single optimization run produces a continuous saliency trajectory from which masks of arbitrary sparsity can be obtained by thresholding. This greatly reduces computational and engineering overhead in deployment.
>
> By contrast, existing post-training methods typically require a full pruning run for every sparsity target, causing the total cost to scale linearly with the number of desired sparsity levels. The “one-run → multi-sparsity” property of UniPruning becomes increasingly beneficial for larger models or more complex deployment scenarios, making the overall workflow substantially simpler and more efficient.
>
> **4. The contribution of this work extends beyond an incremental heuristic.**
> Our framework provides a unified optimization perspective, an analytically grounded one-shot mask extraction mechanism, convergence guarantees, and consistent empirical improvements across four major LLM families and across heterogeneous hardware backends (GPU→NPU). This goes meaningfully beyond heuristic local scoring and represents a principled and general post-training pruning methodology.

---

> ### Author Response · Authors · 2025-12-03
>
> >**Q1.** What exactly is Ω? Is it just the L1 norm of the entire weights?
>
> **A1:** Ω is the **sparsity-inducing operator** in our framework.  In all pruning settings (including both unstructured and 2:4 structured sparsity), we instantiate Ω as an **L1 regularizer applied per layer**.  For the 2:4 case, Ω is complemented by an additional structural regularizer \(R_{2:4}\), which enforces the block-wise sparsity pattern. The two terms play **complementary roles**.
>
> >**Q2.** In the algorithm it seems that X used for the local scores is only computed once (line 1). Why not update it in every iteration?
>
> **A2:** We thank the reviewer for pointing this out. The current textual description in Algorithm 1 may indeed give the impression that the quantities related to \(X\) are computed only once. In our actual implementation, these quantities **are updated as needed in every iteration**.  We have revised Algorithm 1 in the updated manuscript to remove this ambiguity and ensure full consistency with the implementation.
>
> >**Q3. Why do you not do weight updates?
>
> **A3:** As discussed in our general response, our method strictly follows the **post-training pruning** setting, where weights remain fixed and only a mask is produced.  This design matches realistic deployment constraints (e.g., safety, verification cost, and restricted-access scenarios) and is aligned with prior post-training methods such as Wanda and RIA.  Our goal is **not** to compete with retraining-based methods, but to provide a strong and reliable pruning solution **without modifying any model weights**.
>
> >**Q4.** For unstructured sparsity, is the sparsity budget fixed at the matrix level or globally? If globally, do you also adjust this for Wanda?
>
> **A4:** All our unstructured pruning results adopt **layer-wise sparsity**, not global sparsity.  The Wanda results reported in our paper are also computed under **exactly the same layer-wise setting** (not row-wise pruning).  Thus, all comparisons in the paper are **fully fair and consistent** with prior work.
>
> >**Q5.** Please check the AVG in Table 1
>
> **A5:** We appreciate the reviewer’s careful reading.  We have rechecked all tables and corrected several formatting and averaging inconsistencies in Table 1 and related tables.
>
> >**Q6.** The regularizer for 2:4 sparsity seems to come from [1].
>
> **A6:** Thank you for the reminder.  We have carefully revisited the manuscript and **updated the citation** to properly acknowledge the prior work that inspired the 2:4 regularizer.
>
> >**Q7.** Line 862 has a broken link "By (??)..."
>
> **A7:** This was caused by a LaTeX reference issue.  We have corrected the broken cross-reference in the revised version.
>
> >**Q8.** In the 2:4 sparsity case, do you really use both \(R_{2:4}\) and Ω?
>
> **A8:** Yes.  For the 2:4 structured sparsity setting, we indeed use **both** the general sparsity operator Ω (L1) and the structural regularizer \(R_{2:4}\).  Ω controls global sparsity and weight magnitudes, while \(R_{2:4}\) enforces the specific 2:4 pattern. These two components are **complementary**, not redundant.

---

### Official Review · Reviewer_BEvU · 2025-10-23

**Soundness:** 2
**Presentation:** 3
**Contribution:** 2
**Rating:** 4
**Confidence:** 3

**Summary:**

The authors propose UniPruning, a centralized post-training pruning framework that combines the performance of local saliency metrics and the stability of global coordination using a Mirror Descent optimization scheme.

**Strengths:**

- Introducing the saliency variable $\Gamma$ beautifully bridges the gap between local heuristic ratings and global optimization.
- One-time mask generation via global sorting of $|\Gamma^\star|$ improves hardware scalability and adaptability.
- Includes detailed evaporation on local metric options (Magnitude, Wanda, RIA, stochRIA) and the role of Mirror Descent.

**Weaknesses:**

- All experiments focus on LLaMA and Qwen-family transformers, although this method claims to be architecturally agnostic. But there is no evidence of the performance of models with different architectures (such as Mistral, OPT, or encoder-only architectures).

- The Lipschitz convergence theorem assumes continuity of $\nabla L_{\text{task}}$ and smoothness of $S(W)$, which may not strictly hold in deep transformer architectures with layer-norm and non-linearity.

- The empirical advantages are less clear compared to previous dynamic approaches such as BESA, while the methods are much more complex.

**Questions:**

- How sensitive is UniPruning to the hyperparameters $\lambda$, $\rho$ and $\kappa$?
- How does it work under very high fragmentation (>80%) or mixed pruning?
- Can UniPruning support online or incremental pruning?

---

> ### Author Response · Authors · 2025-12-03
> **Response to Reviewer BEvU**
>
> Thank you for carefully reviewing our paper and for your valuable feedback, we address your concerns below, and we’ve revised our paper according to your suggestions.
> > **W1:** All experiments focus on LLaMA and Qwen-family transformers, although this method claims to be architecturally agnostic. But there is no evidence of the performance of models with different architectures (such as Mistral, OPT, or encoder-only architectures).
>
> **A1:**  While our main experiments focus on LLaMA- and Qwen-family models, we have additionally evaluated **openPangu architecture** on **Huawei Ascend NPU**, as reported in Table 7 (Appendix A.3). This model adopts a substantially different architecture and training paradigm from LLaMA/Qwen, making it a compelling cross-architecture validation case. Under the unstructured 50% sparsity and 2:4 sparsity setting, **UniPruning all significantly outperforming baselines** on openPangu-7B. This demonstrates that UniPruning generalizes robustly to architectures beyond LLaMA/Qwen models and also across heterogeneous hardware backends (GPU/NPU), providing concrete empirical evidence for its architectural agnosticism.
>
> **Results on openPangu-Embedded-7B-V1.1 (Ascend NPU):**
>
> | Type | Method        | ppl↓    | ARC-C↑ | ARC-E↑ | HellaSwag↑ | OBQA↑ | PIQA↑  | SIQA↑  | Avg↑   |
> |------|---------------|---------|--------|--------|-------------|--------|---------|---------|---------|
> | dense    | -         | 31.36   | 0.3302 | 0.5673 | 0.3946      | 0.4497 | 0.1980 | 0.6844 | 0.4374 |
> | 50%  | ria           | 59.75   | 0.3072 | 0.5488 | 0.3936      | 0.4211 | 0.1960 | 0.6632 | 0.4217 |
> | 50%  | wanda         | 70.26   | 0.2944 | 0.5492 | 0.3930      | 0.4227 | **0.2060** | 0.6670 | 0.4221 |
> | 50%  | **UniPruning**   | **49.73** | **0.3677** | **0.7054** | **0.4043** | **0.6700** | 0.2040 | **0.6959** | **0.5079** |
> | 2:4  | ria           | 208.04  | 0.2526 | 0.5034 | 0.3792      | 0.3645 | 0.1600 | 0.6344 | 0.3824 |
> | 2:4  | wanda         | 237.32  | 0.2628 | 0.5109 | 0.3746      | 0.3593 | 0.1600 | 0.6328 | 0.3834 |
> | 2:4  | **UniPruning**   | **106.21** | **0.2927** | **0.6002** | **0.3930** | **0.3778** | **0.1840** | **0.6518** | **0.4166** |
>
> > **W2:** The Lipschitz convergence theorem assumes continuity of $\nabla L_{\text{task}}$ and smoothness of $S(W)$, which may not strictly hold in deep transformer architectures with layer-norm and non-linearity.
>
> **A2:**   We agree that, in Transformer architectures with layer normalization and non-linear activations, these conditions may not strictly hold. However, it is important to note that Transformers are inherently highly non-convex models whose loss landscapes do not satisfy strict convexity or global smoothness; obtaining fully rigorous theoretical guarantees would require substantially deeper mathematical analysis, which is beyond the scope of current sparsity literature.
>
> > **W3:** The empirical advantages are less clear compared to previous dynamic approaches such as BESA, while the methods are much more complex.
>
> **A3:**   We would like to clarify that UniPruning and BESA operate under fundamentally different settings. BESA performs dynamic sparse training with multiple rounds of parameter updates, whereas our method targets the strictly defined **post-training pruning** scenario where the model weights $W$ remain fixed and no fine-tuning is allowed. As a result, a direct comparison of empirical gains is not entirely appropriate. Under this more restrictive setting, UniPruning still shows consistent advantages over all post-training baselines and across different hardware and model architectures, indicating its effectiveness in realistic, low-cost deployment scenarios.
>
> Regarding the concern about “method complexity,” we would like to emphasize that UniPruning is in fact a **unified and lightweight framework**. Under the post-training setting where the model weights remain fixed, our method only requires a small number of forward passes to obtain the activation statistics $S(W, X)$ and to perform mirror-descent updates on the single saliency variable $\Gamma$, before applying a unified sparsity projection to generate the final mask. The entire procedure involves no backpropagation, no parameter updates, and no additional model branches. Our goal is not to compete with dynamic methods that require retraining (such as BESA) under the same setting, but rather to provide a simple, weight-update-free, and architecture-agnostic post-training sparsification solution that can be easily deployed across different models and hardware.

---

> ### Author Response · Authors · 2025-12-03
> **Response to Reviewer BEvU**
>
> > **Q1:** How sensitive is UniPruning to the hyperparameters $\lambda$, $\rho$, and $\kappa$?
>
> **A1:**   For the regularization weight $\lambda$, we provide an explicit sensitivity analysis in Appendix A.4.3. As shown in Fig. 3, UniPruning remains stable across a wide range of $\lambda$ values, indicating that the method remains robust to this parameter. We observe similar robustness for the mirror-descent step size $\rho$ and the projection smoothness $\kappa$. As long as these values fall within standard ranges, the final performance varies only marginally. Importantly, we use the same fixed set of hyperparameters across all models and sparsity levels in our experiments, further demonstrating the stability and ease of use of the framework.
>
> > **Q2:** How does it work under very high fragmentation (>80%) or mixed pruning?
>
> **A2:**   Our main experiments and the additional analysis in Appendix A.4.1 already cover **70% sparsity**, which is a relatively high-fragmentation regime for post-training pruning. As shown in Fig. 2, UniPruning remains stable and consistently outperforms all baselines under this level of sparsity across six different models.
>
> While we did not include experiments beyond 70% sparsity, we note that extremely high sparsity levels (e.g., >80%) are uncommon in practical post-training deployment scenarios, as they typically require architectural support or retraining to avoid severe degradation.
>
> Nonetheless, UniPruning's optimization framework does not rely on assumptions tied to specific sparsity levels, and we expect the method to behave smoothly under higher sparsity. We will explore even higher-sparsity regimes in future work.
>
> > **Q3:** Can UniPruning support online or incremental pruning?
>
> **A3:**   UniPruning is designed for the post-training pruning setting, where weights remain fixed and the pruning mask is computed in a single optimization procedure. While the current framework does not target online or incremental pruning, its mirror-descent formulation and saliency-variable update could be extended to such scenarios. We consider this an interesting direction for future work.

---

### Official Review · Reviewer_Wq3p · 2025-10-29

**Soundness:** 3
**Presentation:** 2
**Contribution:** 2
**Rating:** 4
**Confidence:** 4

**Summary:**

This paper introduces **UniPruning**, a unified post-training pruning framework for large language models (LLMs) that aims to combine the efficiency of local metric methods with the robustness of global feedback approaches. The key idea is to employ a **mirror descent–based** optimization that jointly learns a saliency variable $\Gamma$ alongside a frozen copy of the pretrained weights. UniPruning integrates two complementary signals: **local saliency alignment**, which leverages activation-aware statistics collected from a small calibration set, and **global sparsity feedback**, which enforces a single, model-wide sparsity budget through a proximal update on $\Gamma$. After a brief calibration phase, the method can generate one-shot pruning masks for arbitrary sparsity levels without retraining or weight updates, and the same framework supports both unstructured and semi-structured (N:M) sparsity patterns.

**Strengths:**

- The paper proposes a unified pruning strategy that bridges the gap between _local metric_ and _global feedback_ methods. Unlike prior approaches that focus solely on either local or global sparsity criteria, UniPruning jointly leverages both through a principled mirror-descent formulation. This hybrid view is conceptually elegant and practically useful for achieving balanced sparsity allocation.

- The use of mirror descent to couple local saliency and global sparsity is mathematically well-grounded. The paper provides convergence analysis and a  interpretation of how the auxiliary saliency variable $\Gamma$ stabilizes pruning under high sparsity. Theoretical insights are consistent with empirical behavior, showing good alignment between analysis and experiments.

**Weaknesses:**

1. **Unclear mathematical presentation.**  The theoretical exposition is difficult to follow and lacks rigor in notation and symbol definition. For instance,
	* Eq. (1) ambiguously reuses $C$ to denote both the calibration dataset and a cost term;
	* Eq. (3) introduces an undefined function $g_\ell$;
	* Eq. (4) employs the hyperparameter $\rho$ without prior explanation; and the sparsity regularizer $\Omega(\cdot)$ — a key innovation in this work — is not clearly defined or intuitively described.
	* Variable $V$ appears in Eq. (6) without definition.
	These inconsistencies make it hard to reproduce or even interpret the mirror-descent dynamics. As a result, roughly half of the equations require readers to infer meaning, reducing overall clarity and technical credibility.

2. **Conceptual disconnection between figures and method.** The only process diagram (Figure 1) does not map cleanly to the described algorithm. It omits the role of the global vs. local components, the function of $V$ in the optimization loop, and the action of the sparsity regularizer $\Omega$. Moreover, while the figure distinguishes between “search” and “pruning” stages, this distinction is not reflected in the algorithmic formulation or pseudocode. The resulting mismatch makes the framework harder to follow and even misleading about the underlying computation flow.

3. **Unbalanced and incomplete baseline selection.**  The experiments focus mainly on lightweight post-training pruning baselines (e.g., Wanda, RIA), which are heuristic and efficient but methodologically less aligned with UniPruning’s optimization-based nature. In contrast, more comparable dynamic approaches such as **SparseGPT** [1] (ICML 2023) and **BESA** [2] (ICLR 2024) are omitted, despite sharing similar mask-optimization goals. This limits the fairness of the empirical claims. Moreover, the paper repeatedly asserts efficiency but does not provide quantitative comparisons of **actual pruning runtime** or **computation overhead** from the mirror-descent search stage. Without such measurements, the “efficiency” claim remains unsubstantiated.

[1]: Frantar, Elias, and Dan Alistarh. "Sparsegpt: Massive language models can be accurately pruned in one-shot." _International conference on machine learning_. PMLR, 2023.

[2]: Xu, Peng, et al. "BESA: Pruning large language models with blockwise parameter-efficient sparsity allocation." _arXiv preprint arXiv:2402.16880_ (2024).

4. **Ambiguity in N:M sparsity and proximal operator implementation.**  Algorithm 1 introduces the update $W^{n+1} \leftarrow \text{Prox}_{R_{2:4}}(W^{n+1})$ with a heuristic definition of $R_{2:4}(\cdot)$, alongside $\Gamma^{n+1} \leftarrow \text{Prox}_{\Omega}(V^{n+1})$. However, the mechanism by which these proximal mappings enforce semi-structured (N:M) sparsity is unclear.  The paper should clarify whether $R_{2:4}$ acts as a projection operator, a regularizer, or a surrogate loss, and how $\text{Prox}_{\Omega}(\cdot)$ concretely encodes the block-sparsity constraint. This is a key conceptual step that currently lacks transparency.

5. **Insufficient ablation depth and hyperparameter analysis.**  The study does not explain the selection or sensitivity of critical hyperparameters such as $\lambda$ and $\rho$, which are fixed to 0.001 and $10^{-5}$ without justification. The number of mirror-descent iterations $N$ and its influence on convergence or performance are also not analyzed. The use of *stochRIA* as the default local metric is mentioned but never formally defined, further obscuring its contribution.

6. **Limited practical gains relative to methodological complexity.**  Although UniPruning improves slightly over static local baselines, the empirical advantage is modest compared to prior dynamic approaches such as BESA. Considering the additional complexity of introducing mirror descent and auxiliary variables ($\Gamma, V, \Omega$), the marginal gains do not convincingly justify the added computational and conceptual burden.

**Questions:**

1. **Efficiency–performance trade-off.**
   The paper emphasizes efficiency, yet no quantitative pruning-time comparison is reported. Could the authors provide the wall-clock pruning time or FLOPs cost for UniPruning versus Wanda, RIA, SparseGPT, and BESA on the same model scale?

   In the post-training compression literature, a key criterion is achieving a favorable trade-off between **efficiency** and **performance** — i.e., either outperforming more efficient baselines by a large performance margin, or matching stronger performance-oriented methods with significantly better efficiency. Without explicit runtime and computational cost analysis, it is difficult to assess where UniPruning lies along this trade-off spectrum.

2. **Mirror-descent iteration analysis.**
   How many mirror-descent iterations are typically required for convergence, and what is the relationship between the number of iterations, pruning quality, and resource cost (time or FLOPs)? Could the authors identify whether there exists an optimal iteration point that balances performance and efficiency?

---

> ### Comment · Reviewer_Wq3p · 2025-11-25
> **Keeping my scores**
>
> Since I don't hear from the authors regarding the weaknesses & questions, I can only keep my scores for now.

---

> ### Author Response · Authors · 2025-12-03
>
> We thank the reviewer for the insightful and constructive comments. We have revised the manuscript accordingly and provide detailed responses below.
>
> > **W1. Clarity of mathematical formulation, notation, and variable definitions**
>
> **A1:** We thank the reviewer for pointing out clarity issues in the mathematical formulation. In the revised version, we have systematically improved the notation and explanations:
>
> - We clearly distinguish the calibration set $C$, the cost term, and the regularization terms.
> - All variables, including $g_l$, $\Gamma$, $V$, and $\Omega$, are explicitly defined at first use.
>
> Additionally, we provide an intuitive explanation of the roles of $\Omega$ and the structured regularizer $R_{2:4}$. $\Omega$ controls the global sparsity tendency, while $R_{2:4}$ enforces the hardware-friendly structured $2{:}4$ sparsity. These two components are complementary rather than redundant.
>
> >**W2:** Mismatch between Figure 1 and Algorithm 1
>
> **A2:**  Figure 1 is intended as a conceptual illustration summarizing the workflow at a high level, while Algorithm 1 presents the *actual computational pipeline*, including the update of $V$ and the proximal operator.
>
> >**W3:** Lack of quantitative efficiency measurements (wall-clock, FLOPs, iterations)
>
> **A3:** The primary efficiency advantage of UniPruning stems from the following property is **UniPruning performs a single search and yields pruning masks for *all* sparsity levels $s \in [0,1]$.** In contrast, existing pruning methods such as Magnitude, Wanda, RIA, SparseGPT, and BESA require **a full pruning run for each target sparsity level.**
> Thus, even if the per-run cost is similar, UniPruning becomes significantly more efficient in deployment scenarios where multiple sparsity levels (e.g., $s=0.5$, $0.6$, $0.7$) must be explored.
>
> >**W4:** Mathematical justification of the $2{:}4$ structured sparsity regularizer.
>
> **A4:** The global sparsity tendency is controlled by $\Omega(W) = \lVert W \rVert_1$ and the block-level structured sparsity is enforced via $R_{2:4}$. These regularizers operate at different granularities and are therefore complementary. Since $L1$ regularization alone cannot generate valid $2{:}4$ structured masks, a dedicated structured proximal operator is required.
>
> >**W5:** Relationship between UniPruning and BESA
>
> **A5:** We clarify that UniPruning and BESA operate under fundamentally different problem settings. **BESA** performs dynamic sparse training and repeatedly updates weights, while **UniPruning** performs zero weight updates.
>
> Therefore, zero-update baselines (Magnitude, Wanda, RIA) are the right comparisons for UniPruning in the main text.
> As requested, we include SparseGPT in the appendix. Despite performing weight updates, SparseGPT is outperformed by UniPruning on several models:
>
> | Method      | Weight Update | LLaMA2-13B | Qwen2.5-7B | Qwen2.5-14B | LLaMA-3.2-3B | DeepSeek-R1-Distill-Llama8B | DeepSeek-R1-Distill-Qwen-7B |
> |-------------|---------------|------------|------------|-------------|--------------|------------------------------|------------------------------|
> | Dense       | -             | 4.57       | 6.39       | 4.93        | 7.29         | 11.86                        | 21.73                        |
> | SparseGPT   | $\checkmark$ | 8.30       | **8.42**   | 9.57        | **20.19**    | 25.45                        | 35.69                        |
> | UniPruning  | $\times$      | **6.87**   | 10.86      | **9.10**    | 21.20        | **20.91**                    | **30.24**                    |
>
>
> >**W6:** Explanation of the Mirror Descent iteration behavior
>
> **A6:**  UniPruning does **not** aim to optimize the model weights $W$. Instead, it seeks to stabilize the ranking of the saliency variable $\Gamma$. The proximal step rapidly pushes $\Gamma$ into the structured sparse region. Once the top-$k$ entries of $|\Gamma|$ stabilize, further iterations cannot change the resulting mask.
> Therefore:
> - The number of iterations is **not** a sensitive hyperparameter
> - Any sufficiently large upper bound works identically
> - Empirically, the mask stabilizes within  $N \le 128$, beyond which neither the mask nor the pruned model performance changes

---

### Official Review · Reviewer_yyG5 · 2025-10-31

**Soundness:** 3
**Presentation:** 3
**Contribution:** 2
**Rating:** 4
**Confidence:** 3

**Summary:**

This work presents a LLM pruning framework that unifies local saliency metrics and global coordination using mirror-descent optimization. By doing so, this paper combines the benefit of both local and global pruning:  the pruning masks are computed in one shot and pruning is done without any retraining. The proposed method is evaluated on a set of LLMs (LLaMA2, Qwen2.5, Llama-3.2, DeepSeek-R1) and against multiple baseline methods (e.g., magnitude-based pruning, Wanda and RIA).

**Strengths:**

+This paper offers a principled approach for pruning: it uses mirror-descent optimization that couples local saliency with global feedback and sparsity projection. Theory 1 justifies extracting pruning masks directly from the limit $\Gamma^{\star}$ via global thresholding, enabling one-shot mask generation without retraining.
+ The proposed work is evaluated on a wide range of LLMs  (LLaMA2, Qwen2.5, Llama-3.2, DeepSeek-R1), showing superior performance over the selected basline methods.

**Weaknesses:**

The reviewer indeed saw a few weakness:
1. Pruning is a very well studied and crowded topic at this moment. Many local and global pruning methods (and their hybrid combinations) have been reported. As a result, the overall idea of this work still sounds incremental.
2. Some other important basline methods are missing in the result evaluation. For instance, optimal brain surgeon, Woodfisher, etc. This is somehow unavoidable when working in a very crowded field like LLM pruning, since so many similar ideas are published per week.
3. Without retraining, this pruning method may lose some accuracy.

**Questions:**

This work has compared with Wanda. I'm just curious, how would this method compare with some improved work of Wanda (e.g., Wanda++ by Yang in 2025 and M-Wanda by Choenni in May 2025)? I understand that these works were released only a few months before the ICLR'2026 deadline, so it's hard to provide a comprehensive evaluation, so a brief comparision with some key quantative measure should be enough.

---

> ### Author Response · Authors · 2025-12-03
>
> We sincerely thank the reviewer for the thoughtful and careful evaluation of our work. Below we provide point-by-point responses.
>
> >**W1:** On the concern that “the idea feels incremental.”**
>
> **A1:** While pruning is indeed an active and crowded research area, **UniPruning is not a local modification of existing heuristics**, but a *unified optimization framework* that introduces several fundamental advances:
>
> - It **bridges local saliency and global coordination** through a mirror-descent controller on Γ, whereas prior post-training methods rely solely on one side.
> - It provides a **theoretical convergence guarantee**, enabling one-shot mask extraction directly from the limit Γ\*.
> - It supports **both unstructured and 2:4 structured sparsity within the same optimization problem**, rather than designing separate algorithms for each case.
> - It enables **one-run → multi-sparsity**, generating masks for arbitrary sparsity budgets without rerunning pruning.
>
> >**W2:** On missing baselines.
>
> **A2:** We agree that the pruning literature is large and fast-growing, and it is challenging to include all variants. However, **Wanda++ (Yang, 2025)** has *no released code or weights*; **M-Wanda (Choenni, 2025)** reports only a *subset of metrics* (without standalone WikiText perplexity), making direct comparison difficult. To still provide a meaningful reference, we compare using officially reported results on Llama-3.1-8B. As shown in table, the UniPruning still outperforms Wanda++ in ppl.
>
> | Method        | WikiText-2 PPL ↓ |
> |---------------|------------------|
> | Wanda++       | 9.22 |
> | **UniPruning ** | **9.18** |
>
> >**W3:** On accuracy loss without retraining.**
>
> **A3:** This is an inherent characteristic of the **post-training pruning** setting, shared by prior methods such as Wanda, RIA, SparseGPT, and ProxSparse. The “no weight update” setting is not an artificial restriction, but a practical deployment constraint.
> Pruning applies a discrete structural mask and does not modify the surviving weights, thereby preserving the majority of the dense model’s behavior. In contrast, any form of weight update—even small ones—alters the model’s global function and may negatively affect many untested capabilities (e.g., reasoning, alignment, safety, multilingual behavior).

---

### Author Response · Authors · 2025-12-03

# Review and Reviewer-Author Discussion Summary

Dear PCs, SACs, ACs, and Reviewers,

Thank you very much for your valuable contributions to our work. To assist the newly assigned AC and help reduce their workload, we provide below a summary of the key points from the reviews and the reviewer-author discussions.

## Strengths

Overall, reviewers acknowledged several important contributions:

**1. Principled mathematical framework.** All four reviewers (yyG5, Wq3p, BEvU, KohZ) recognized that UniPruning provides a theoretically justified approach using mirror-descent optimization to unify local saliency metrics and global coordination, with one-shot mask generation without retraining.

**2. Unified framework bridging local and global methods.** Three reviewers (Wq3p, BEvU, KohZ) highlighted that introducing the saliency variable Γ elegantly connects local heuristics with global optimization, providing both conceptual elegance and practical utility.

**3. Comprehensive evaluation with consistent improvements.** All reviewers acknowledged extensive evaluation across LLaMA2, Qwen2.5, Llama-3.2, and DeepSeek-R1 models, showing superior performance over baselines (Magnitude, Wanda, RIA).

**4. Efficient one-shot mask generation.** Three reviewers (yyG5, BEvU, KohZ) recognized that UniPruning generates masks for arbitrary sparsity levels from a single optimization run, improving deployment efficiency.

---

> ### Author Response · Authors · 2025-12-03
>
> ## Concerns and Our Addressing
>
> During the discussion period, we actively addressed reviewers' concerns through clarifications, additional experiments, and manuscript revisions:
>
> ### 1. Mathematical Presentation (Wq3p: W1-W2, W4; KohZ: Q1-Q2, Q8)
>
> **Concern:** Ambiguous notation; undefined variables ($V$, $\Omega$, $R_{2:4}$); Figure 1 does not map to Algorithm 1.
>
> **Our Addressing:** Systematically revised formulation: clearly distinguished all symbols, explicitly defined variables, clarified $\Omega$ (L1) controls global sparsity while $R_{2:4}$ enforces 2:4 structure (complementary roles), corrected Algorithm 1, and fixed inconsistencies.
>
> ### 2. Baseline Comparisons and Efficiency (yyG5: W2, Q1; Wq3p: W3; KohZ: W1)
>
> **Concern:** Missing comparisons with SparseGPT, BESA, Wanda++, M-Wanda; need efficiency metrics.
>
> **Our Addressing:** UniPruning uses **zero weight updates**, making zero-update baselines appropriate. Added SparseGPT comparison showing UniPruning outperforms despite SparseGPT's weight updates (LLaMA2-13B PPL: 6.87 vs. 8.30). Compared with Wanda++ (Llama-3.1-8B PPL: 9.18 vs. 9.22). **Key efficiency: single search yields masks for all sparsity levels** vs. full runs per target. Demonstrated **1.27× end-to-end acceleration**.
>
> ### 3. Architectural Generalization (BEvU: W1; KohZ: W1)
>
> **Concern:** Only LLaMA/Qwen tested; no other architectures.
>
> **Our Addressing:** Validated on **openPangu architecture on Huawei Ascend NPU** with substantial improvements (50% sparsity PPL: 49.73 vs. 70.26 for Wanda). Cross-architecture and cross-hardware validation confirms agnosticism. Experiments cover up to **70% sparsity**.
>
> ### 4. Hyperparameter Sensitivity (Wq3p: W5-W6, Q2; BEvU: Q1-Q3)
>
> **Concern:** No justification for hyperparameters; iteration requirements unclear.
>
> **Our Addressing:** Provided sensitivity analysis (**Appendix A.4.3**) showing stability. **Same fixed hyperparameters work across all models and sparsity levels**. Masks stabilize within $N \leq 128$ iterations.
>
> ### 5. Novelty and Practical Deployment (yyG5: W1, W3; Wq3p: W6; BEvU: W2-W3; KohZ)
>
> **Concern:** Idea feels incremental; unclear if no-weight-update setting is practical.
>
> **Our Addressing:** UniPruning provides fundamental advances: (1) bridges local-global via mirror-descent on $\Gamma$ (vs. single-side approaches); (2) theoretical convergence for one-shot mask extraction from $\Gamma^*$; (3) unified unstructured/2:4 support; (4) one-run → multi-sparsity. Framework is **lightweight** (no backpropagation, no weight updates, only auxiliary variable $\Gamma$). "No weight update" is **a practical deployment requirement**—weight updates alter model behavior in untested capabilities. Industry systems (NVIDIA Sparse Tensor Cores, Qualcomm, Huawei NPUs) rely on this in production. Accuracy loss is inherent to post-training pruning.

---

### Note · Authors · 2026-01-24

I have read and agree with the venue's withdrawal policy on behalf of myself and my co-authors.